# Asking Before Acting: Gather Information in Embodied Decision-Making with Language Models

## Abstract

With strong capabilities of reasoning and a broad understanding of the world, Large Language Models (LLMs) have demonstrated immense potential in building versatile embodied decision-making agents capable of executing a wide array of tasks. Nevertheless, when deployed in unfamiliar environments, we show that LLM agents encounter challenges in efficiently gathering essential information, leading to suboptimal performance. Conversely, human individuals often seek additional information from their peers prior to taking action, harnessing external knowledge to avoid unnecessary trial and error. Drawing inspiration from this behavior, we propose *Asking Before Acting* (ABA), a method that empowers the agent to proactively inquire with external sources for pertinent information using natural language during their interactions within the environment. In this way, the agent is able to enhance its efficiency and performance by circumventing potentially laborious steps and combating the difficulties associated with exploration in unfamiliar environments and vagueness of the instructions. We conduct extensive experiments involving a spectrum of environments including text-based household everyday tasks, robot arm manipulation tasks, and real world open domain image based embodied tasks. The experiments involve various models from Vicuna to GPT-4. The results demonstrate that, even with modest prompts modifications, ABA exhibits substantial advantages on both performance and efficiency over baseline LLM agents. Further finetuning ABA with reformulated metadata (ABA-FT) faciliates learning the rationale for asking and allows for additional enhancements especially in tasks that baselines struggle to solve.

## 1 Introduction

Recent advances in Large Language Models (LLMs) have exhibited remarkable abilities in language comprehension, text generation, question answering, dialogue, reasoning, and can even exhibit human-level performance on various benchmarks (Ouyang et al., 2022; Chowdhery et al., 2022; OpenAI, 2023; Google, 2023). Since LLMs have been trained on extensive and diverse text corpora, they have captured a broad range of commonsense understanding about the world, enabling them to adeptly handle a multitude of various and complex scenarios. Therefore, recently, researchers have proposed to integrate LLMs in embodied decision-making (Huang et al., 2022a; Li et al., 2022; Ahn et al., 2022; Huang et al., 2022b; Singh et al., 2022a; Yao et al., 2022; Wang et al., 2023; Driess et al., 2023; Carta et al., 2023), using LLMs either to do task planning or as autonomous agent.

LLM agents exhibit strong decision-making capabilities when presented with comprehensive information, yet they may encounter difficulties when confronted with scenarios characterized by limited or insufficient information, such as with underspecified instructions, or in unfamiliar environments. In such cases, the agents may struggle to make informed decisions. To illustrate this, consider the scenario depicted in Figure 1, where a robot is deployed in an office and tasked with packing items. As shown in Figure 1 (c), act without asking may lead to suboptimal behaviors. The vagueness present in the instruction leave the true reward function underspecified, and an arbitrary mapping from instruction to actions might not yield desirable outcomes. Furthermore, when the robot lacks specific prior knowledge like an object's location, directly taking actions may result in a systematic search strategy. Even though the robot may finally manages to locate the target object, this whole

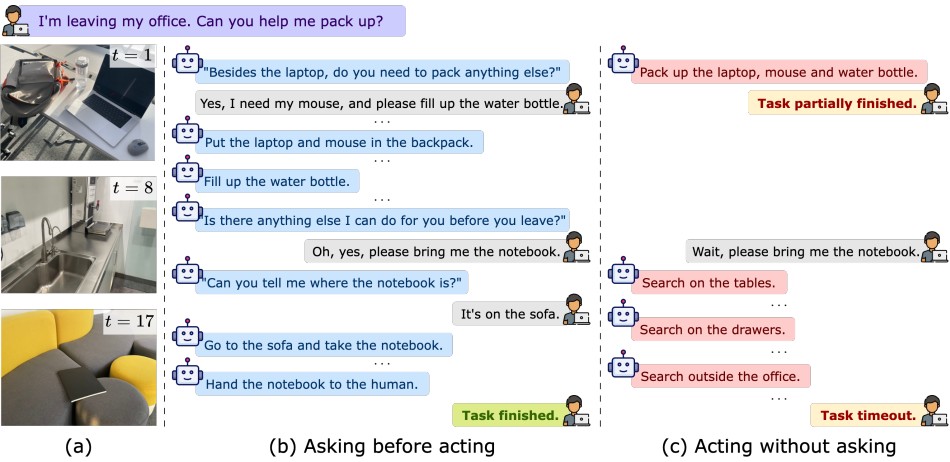

Figure 1: ABA (as shown in (b)) allows the agent to seek clarification about the task as well as efficiently gather necessary information. In contrast, acting without asking (as showin in (c)) may lead to task misunderstanding or laborious exploration perid. Note (a) and (b) are key frames of trajectories generated by ABA in experiments in Section 4.3.

process is conspicuously inefficient, let alone the possibility of suboptimal searching behavior which may lead to failure.

In contrast, when confronted with such scenarios, we humans typically adopt a different approach. When assigned with ambiguous tasks, it is more prudent to ask for clarification about the task before taking actions, ensuring we have the tasks correctly understood. Besides, rather than resorting to onerous trial and error, it is natural for us to actively query external information from our peers to accelerate information gathering and guide decision making. As shown in Figure 1 (a) and (b), we excerpt key frames and logs produced by our method (refer to Section 4.3 for details). Our policy first seeks clarification about the task by inquiring about the human's preferences. Then, instead of searching every hole and corner, the agent opts to ask "Can you tell me where the notebook is?" and directly proceeds to that location. The comparision builds us an intuition that, by posing a few questions in natural language, the agent can enjoy an substantial enhancement in both policy performance and efficiency.

Building upon this intuition, we focus on a novel setting where the agent can actively query for additional information from external sources using natural language during their interactions with environments. Though some existing works have explored scenarios involving human-in-the-loop interactions to provide additional information, our setting is stands apart from these previous ones. A majority of works (Nguyen & Daumé III, 2019; Nguyen et al., 2019; Singh et al., 2022b; Da Silva et al., 2020) ask humans for oracle actions or action descriptions, Nguyen et al. (2022) ask for information about current states and (sub-)goals. Liu et al. (2022) asks three-word-templated questions to accelerate training, while Huang et al. (2022b) ask for scene, task, or preferences descriptions. In contrast to existing works, our setting concentrates on designing a generic mechanism to gather information through natural language, which imposes fewer restrictions and aligns more closely with human decision-making processes.

In this paper, we begin by introducing a novel theoretical framework termed *Contextual MDP with Human / External Information Sources in the Loop* in Section 3.1, which integrates the information querying into decision making process by incorporating human in the formulation. In Section 3.2, we propose *Asking Before Acting* (ABA), an efficient method that is able to accelerate information gathering by actively querying in natural language while interacting with the environments. ABA can learn to ask proper open-domain questions and effectively leverage the gained information only with modest modifications on prompts and use pretrained (visual) language models out of the box. To further improve the performance, we introduce ABA-FineTuning (ABA-FT). We propose to reformulate the metadata associated with the question formulation as question-answering process, which aids the model in understanding the rationale behind posing questions. Besides, we also introduce a simple yet effective evluation method in Section 3.3.

To evaluate the capability of ABA and ABA-FT, we conduct extensive experiments in Section 4, involving a spectrum of environments including text-based household everyday tasks, robot arm manipulation tasks, and real world open domain image based embodied tasks. We also evaluate various models encompassing Vicuna (Chiang et al., 2023), GPT-3.5 and GPT-4 to test the scalbility of

our method. The quantitative results demonstrate that, even with modest modifications on prompts, ABA obtains consistent performance gains across almost all environments against baseline LLM agents and classical imitation learning method. Further finetuning with a fraction of data enables ABA-FT to achieve additional enhancements, especially in tasks that baselines struggle to solve. To better understand the behavior and practical value of ABA, we conduct experiments on real world open domains embodied tasks with image inputs in Section 4.3. The qualitative results provide additional evidence that, when combined with a versatile controller, ABA is capable of formulating open-domain questions grounded in the context of image observation to gather information. This showcases the practical value of ABA and augurs a promising direction for future research.

## 2 PRELIMINARIES

To better formulate the information gathering procedure, we first introduce Contextual Markov Decision Processes (Contextual MDPs) (Hallak et al., 2015):

**Definition 2.1** *Contextual MDP is a tuple* $(\mathcal{S}, \mathcal{A}, \mathcal{C}, \mathcal{M}(c))$. *Here* $\mathcal{S}$ *and* $\mathcal{A}$ *stand for state space and action space respectively.* $\mathcal{C}$ *is the context space.* $\mathcal{M}$ *is a function mapping context* $c \in \mathcal{C}$ *to a specific* $T$-*horizon MDP* $\mathcal{M}(c) = (\mathcal{S}, \mathcal{A}, p(\cdot|s, a, c), r(s, a, c))$.

Here, $\{\mathcal{M}(c), \forall c \in \mathcal{C}\}$ represents a family of MDPs characterized by a shared state space $\mathcal{S}$ and action space $\mathcal{A}$, but with different transition function $p(\cdot|s, a, c)$ and reward function $r(s, a, c)$ specified by $c$. The goal of the agent is to learn a policy $\pi$ to maximize the accumulative rewards on the target environment(s). Note that the context $c$ varies across different environments, and oftentimes, it remains unknown. Optionally, the agent will be additionally provided with a task instruction $i$, which is usually a concise language description of the goal, providing extra information about the context $c$. As shown in Defenition 2.1, when deployed to a new environment, understanding the context $c$ becomes a prerequisite for achieving high rewards since it indicating the transition and reward functions. In light of this, one common approach is to gather information about the context $c$ through interactions with the environment by trial and error, i.e., infer the context from history $\hat{c} = f_\theta(i, s_1, a_1, r_1, \cdots, s_t)$ while trying to solve the task. However, efficiently gathering information in various unknown environments with different contexts $c$ can be challenging. Aside from limited generalization capability Beck et al. (2023), existing methods often rely on dense rewards and sufficiently small state space (Zintgraf et al., 2021), which may lead to catastrophic failure in embodied decision-making where the environments often lack carefully crafted dense reward functions and the state spaces are often large.

We argue that this is not, at least always, the case how humans deal with the task. Instead of attempting to figure out $c$ solely on our own, we typically seek clarification of tasks or assistance from another human, often a more experienced senior, to obtain valuable information. This behavior can significantly alleviate the exploration burden for $c$ in many situations. The above insight urges us to reconsider the process of embodied decision-making in unfamiliar evaluation environments: What if the agent does not necessarily need to figure out everything on its own? While certain prior research has studied scenarios involving human-in-the-loop interactions, our setting concentrates on the development of a generic mechanism with fewer restrictions and better alignments with human behavior (see Section 5 for a detailed survey) and we are the pioneers in addressing the challenge of enabling information gathering for embodied decision-making with LLM agents.

## 3 METHOD

In this section, we present our new problem formulation as well as the our algorithm.

### 3.1 CONTEXTUAL MDP WITH HUMAN / EXTERNAL INFORMATION SOURCE IN THE LOOP

To incorporate humans (or other external knowledge sources) in the loop of decision-making, we opt to integrate the physical interactions with environment and language interactions with humans as a unified interface:

**Definition 3.1** *Contextual MDP with Human / External information source in the loop based on* $(\mathcal{S}^U, \mathcal{A}^U, \mathcal{C}, \mathcal{H}(c), \mathcal{M}(c))$. *Here* $\mathcal{S}^U, \mathcal{A}^U$ *are the augmented state space and action space:* $\mathcal{S}^U = \mathcal{S} \cup \mathcal{L}_{ans}, \mathcal{A}^U = \mathcal{A} \cup \mathcal{L}_{ask}$, *where* $\mathcal{L}_{ask}$ *and* $\mathcal{L}_{ans}$ *include all possible questions and answers in natural language.* $\mathcal{H}(c)$ *maps context* $c \in \mathcal{C}$ *to* $\mathcal{H}_c$, *which is a model of human (or other external information source) in context* $c$ *that can map any questions to information.* $\mathcal{M}(c) = (\mathcal{S}^U, \mathcal{A}^U, \mathcal{H}_c, p_U(\cdot|s, a, c, \mathcal{H}_c), r(s, a, c), \gamma)$

Like Contextual MDP, $\mathcal{M}(c) = (\mathcal{S}^U, \mathcal{A}^U, \mathcal{H}_c, p_U(\cdot|s,a,c,\mathcal{H}_c), r(s,a,c), \gamma)$ has a transition function and a reward function parameterized by $c$. However, the state space $\mathcal{S}^U$ and the action space $\mathcal{A}^U$ are augmented with answers and questions respectively, and the transition function $p_U(\cdot|s,a,c,\mathcal{H}_c)$ can be factorized as:

$$p_U(s'|s,a,c,\mathcal{H}_c) = p(s'|s,a,c) \cdot \mathbb{1}_{a \in \mathcal{A}} + p(\mathcal{H}_c(a) = s') \cdot \mathbb{1}_{a \in \mathcal{L}_{ask}} \tag{1}$$

With the augmented action space, the agent can now query to gather information while interacting with the environments. For instance, by simply asking "where is the kitchen?", the agent can omit tens of steps of exploration to find the kitchen. However, several challenges exist: First, when deployed to unfamiliar environments, it is important for the agent to identify the key information that is pertinent while filtering out the task-irrelevant ones. Sencondly, to better communicate with humans, the agent should do reasoning to map between abstract natural language and specific object in observation. This requires both grounded question proposing as well as grounded answer understanding. Furthermore, it is also desirable to ask only when necessary, i.e., when it cannot reason the answers from historical information. To solve these challenges, we propose *Asking Before Acting* (ABA), an effective method for the language agent to cleverly gather necessary information.

## 3.2 ALGORITHM

In this paper, we build our model based on pretrained (visual) language models, and we focus on the setting where the task instruction $i$ is provided . We define $\tau_t = \texttt{concat}(i, s_1, a_1, \cdots, s_t)$, which is the concatenation of the instruction and the historical observations and actions. We describe our methods in the following sections:

### 3.2.1 ASKING BEFORE ACTING

In-Context Learning (ICL) (Brown et al., 2020) can approximate a function approximator $\hat{f}$ from sequences of pairs $(x_i, f(x_i))_i$ and allow to predictions $\hat{f}(x')$ on novel $x'$. Recently, ICL draws a lot of attention (Xie et al., 2021; Akyürek et al., 2022; Dai et al., 2022) due to its superior efficiency and the ability of generalization. Recent works on embodied planning (Huang et al., 2022a; Singh et al., 2022a) propose use ICL to learn to output actions to interact with the environment given $K$ trajectories $\tau^k = \texttt{concat}(i^k, s_1^k, a_1^k, \cdots, s_T^k, a_T^k)$, where $k \in \{1, 2, \cdots, K\}$ . However, directly applying ICL may result in limited generalization and poor performance (Yao et al., 2022). Yao et al. (2022); Ahn et al. (2022) propose to model the reasoning steps of actions via the Chain-of-Though prompting (Wei et al., 2022) by explicitly expanding $\tau^k$ to incorporate a "think" step.

Based on Yao et al. (2022); Ahn et al. (2022), we propose to learn the rationale for question asking via ICL. Suppose there is a question at time $t$. Rather than directly mapping $\tau_t$ to the questions $a_t$, we identify the corresponding context $c_t$ the question $a_t$ is querying for ($c_t$ may correspond to a partial of $c$). We term the identification procedure, i.e., why we posing such questions, as the metadata of the questions. For instance, let $a_t$ be a question asking "where is the notebook". Instead of directly learning to ask, we insert the metadata of $a_t$, which we denote as $\bar{a}_t$: "to finish the task, we need to find the notebook. But I do not know its location, so I need to ask for it". Since $\bar{a}_t$ and $a_t$ are all manually labelled, it reveals the rationale for asking $a_t$ of the labeller. By providing $K$ such trajectories, we learn through ICL to map from $\tau_t$ to human-like analysis of what context is desirable yet unknown right now (if so), and from the analysis to human-like questions. To this end, ABA aligns the agent with human asking habits. Denote $\bar{\tau}^k$ as the augmented trajectories with $\bar{a}_t$ prepend before each question $a_t$, and $\tilde{\mathcal{A}}$ to be the augmented action space with $\mathcal{L}_{ask}$ augment to concatenation of $[\mathcal{L}_{meta}, \mathcal{L}_{ask}]$, the action is selcted by:

$$\tilde{a}_t = \arg\max_{\tilde{a} \in \tilde{\mathcal{A}}} \prod_{i=0}^{|a|} \pi_{LLM}(e_i|\bar{\tau}^1, \cdots, \bar{\tau}^K, \tau_t, e_{1:i-1}) \tag{2}$$

where $e_i$ is the $i$-th token of the action, and $|a|$ refers to the number of tokens. Note $\bar{a}_t$ is null when $a_t$ is not a question. We can derive final action $a_t$ by dropping the potential $\bar{a}_t$ in $\tilde{a}_t$ if needed.

### 3.2.2 ASKING BEFORE ACTING WITH FINETUNING

In experiments, we find that ICL excels at scenarios when the the task time horizon is relatively short. As the task horizon increases, due to the token limitations of (visual) language models, sometimes we can only use 1 example which results in limited diversity and thus hampers the performance of ICL. However, longer horizon usually corresponds to more complex policy which exacerbates the need for samples. Therefore, a more proper treatment is needed especially when the horizon is long.

To this end, we propose ABA with finetuning (ABA-FT). We first describe how we collect two datasets $\mathcal{D}_{QA}$ and $\mathcal{D}_{\pi}$. We label $N$ trajectories with expert policy where each trajectory consists of input-output pairs for $T_i$ timesteps as $\{(\bar{\tau}^i_t, \bar{a}^i_t, a^i_t, n^i_t)_{t=0}^{T_i}\}_{i=0}^{N}$. To alleviate the distribution shift problem (Ross et al., 2011), we intentionally corrupt the expert policy by randomly injecting noisy actions with probability $p$ and mark this as noise by setting $n^i_t = 1$ in the dataset. Then, we can reformulate these trajectories as two datasets.

For $\mathcal{D}_{QA}$, we convert the dataset to the form of question-answering for each $\bar{a}_t$. For instance, when looking for a notebook, we provide the agent with $\bar{\tau}_t$ and ask "do you have ever seen the notebook" and label the answer to "yes" nor "no" to indicate whether it is necessary to ask. We also include the question "where have you seen the notebook" and its corresponding answer if previous answer is yes. In this way, we hope the agent can learn to remember the key factors of inferring the rationale for asking. As for the policy dataset $\mathcal{D}_{\pi}$, we do a modified version of imitation learning by masking out all the corrupted actions, since we want the agent to learn to recover from mistakes and we do not want it to learn to take wrong actions intentionally. For notation simplicity, we denote $\mathcal{D}_{QA} = \{(x^i_{QA}, y^i_{QA})\}_{i=0}^{N_\pi}$, where $x^i_{QA}$ corresponds to the inputs as well as questions, and $y^i_{QA}$ corresponds to the answer. We also denote $\mathcal{D}_{\pi} = \{(x^i_{\pi}, y^i_{\pi}, n_i)\}_{i=0}^{N_\pi}$ where $x^i_{\pi} = \tau^j_t$ and $y^i_{\pi} = [\bar{a}^j_t, a^j_t]$ for $j \in \{0, \cdots, N\}$. To conclude, ABA-FT is trained to maximize the following objectives:

$$\mathcal{L} = -\sum_{i=0}^{N_{QA}} \log \pi_{LLM}(y^i_{QA}|x^i_{QA}) - \sum_{i=0}^{N_\pi} \log \pi_{LLM}(y^i_{\pi}|x^i_{\pi}) \cdot \mathbb{1}_{n_i=0} \tag{3}$$

### 3.3 Human Model

To minimize human involvement, during the evaluation phase, we propose a simple yet effective method for open-domain human in the loop problems. We implement the model of human (or other information sources) $\mathcal{H}$ via another language model which is instructed to respond to questions based on the provided information. To incorporate prior knowledge about the environment, we extract the information about the object placement from the simulator and transform it into a descriptive paragraph. This paragraph is then fed to $\mathcal{H}$. Whenever the agent poses a question, $\mathcal{H}$ is tasked with providing an answer based on the paragraph. It's worth noting that this design allows for the straightforward replacement of the current language model with human or alternative information sources for more appropriate answers. For more details, please refer to Appendix A.

## 4 Experiments

In this section, we empirically evaluate ABA on a series of embodied decision-making tasks.

In Section 4.1, we evaluate our algorithm in three household embodied decision-making scenarios, namely ALFWorld (Shridhar et al., 2021) and its variants, to demonstrate its capability to handle high level embodied tasks. In Section 4.2, we extend our evaluation to control tasks to demonstrate that ABA is able to build mappings between language questiosn and low level control tasks. Finally, in Section 4.3, to further prove the effectiveness of ABA, we build 4 real world scenarios. We also replace the human model by real humans to answer the questions and communicate with ABA. The qualitative results showcase the practical value of ABA and augurs a promising direction for future research.

### 4.1 Text-based Household Everyday Task

**ALFWorld** ALFWorld (Shridhar et al., 2021) is an interactive text-based environment with aligned embodied worlds, which contains six types of different everyday household tasks. As for baselines, we select two representative baselines, BUTLER (Shridhar et al., 2021) and ReAct (Yao et al., 2022). BUTLER (Shridhar et al., 2021) is an imitation learning based method without LLM. It trains an independent model for each task, while each of the model uses a substantial dataset of $10^5$ expert trajectories. For ReAct and ABA, we use $K = 2$ in-context examples. For more details, please refer to Appendix B.

The results are presented in Table 1 and 2. The results of BUTLER are directly taken from Shridhar et al. (2021) which reports the best performance across 8 different seeds. For ReAct and ABA, we report the performance mean and standard deviation across 5 seeds for Vicuna-7B (Chiang et al.,

Table 1: Success rate on ALFWorld environments. `ID` and `OOD` refer to in distribution and out-of-distribution evaluation set provided in Shridhar et al. (2020). `V7B` and `G35` refers to Vicuna 7B and GPT-3.5 API. Note that for BUTLER, we report the best success rates across 8 seeds to align with the original paper, while for ReAct and ABA, we report mean success rates across 5 or 3 seeds.

|  |  | Pick | Examine | Clean | Heat | Cool | Pick 2 | All |
|---|---|---|---|---|---|---|---|---|
| BUTLER | ID | 61 | 39 | 44 | **81** | 60 | 29 | 40 |
| max of 8 | OOD | 46 | 22 | 39 | 74 | **100** | 24 | 37 |
| ReAct / V7B | ID | $9 \pm 7$ | $8 \pm 4$ | $9 \pm 3$ | $4 \pm 5$ | $9 \pm 8$ | $4 \pm 4$ | $7 \pm 3$ |
| mean of 5 | OOD | $3 \pm 3$ | $6 \pm 3$ | $5 \pm 3$ | $10 \pm 3$ | $2 \pm 3$ | $9 \pm 5$ | $6 \pm 1$ |
| ABA / V7B | ID | $\mathbf{60 \pm 6}$ | $\mathbf{52 \pm 5}$ | $\mathbf{59 \pm 6}$ | $46 \pm 6$ | $\mathbf{61 \pm 3}$ | $\mathbf{61 \pm 10}$ | $\mathbf{56 \pm 3}$ |
| mean of 5 | OOD | $37 \pm 5$ | $\mathbf{53 \pm 5}$ | $51 \pm 2$ | $52 \pm 6$ | $50 \pm 15$ | $\mathbf{41 \pm 0}$ | $\mathbf{48 \pm 2}$ |
| ReAct / G35 | ID | $62 \pm 4$ | $67 \pm 1$ | $67 \pm 3$ | $67 \pm 2$ | $55 \pm 1$ | $67 \pm 1$ | $64 \pm 1$ |
| mean of 3 | OOD | $67 \pm 13$ | $67 \pm 2$ | $75 \pm 10$ | $67 \pm 7$ | $54 \pm 8$ | $59 \pm 0$ | $65 \pm 2$ |
| ABA / G35 | ID | $\mathbf{86 \pm 6}$ | $\mathbf{79 \pm 7}$ | $\mathbf{79 \pm 2}$ | $73 \pm 4$ | $\mathbf{81 \pm 11}$ | $\mathbf{98 \pm 2}$ | $\mathbf{84 \pm 3}$ |
| mean of 3 | OOD | $\mathbf{83 \pm 4}$ | $\mathbf{78 \pm 6}$ | $\mathbf{92 \pm 2}$ | $\mathbf{81 \pm 14}$ | $\mathbf{85 \pm 6}$ | $\mathbf{84 \pm 7}$ | $\mathbf{85 \pm 4}$ |

2023) and 3 seeds for GPT-3.5 (Ouyang et al., 2022) API. The empirical results demonstrate that ABA strongly outperforms other baselines, and also shows great efficiency in terms of parameters.

As shown in Tabel 1, with one unified model, ABA can successfully handle most scenarios. The mean performance of ABA with GPT-3.5 is higher than the max performance of BUTLER in 12 out of 14 scenarios, using only $K = 2$ in-context examples and one model versus $10^5$ expert trajectories and separate training according to BUTLER. ABA also outperforms ReAct in all scenarios and better efficiency in terms of model parameters. When deployed with a relatively smaller model such as Vicuna-7B, ABA achieves more than 40% performance gain compared to ReAct. This proves that our method can be efficiently deployed to scenarios with limited computational resources. When the backbone is switched to GPT-3.5, both ReAct and ABA gain performance improvements, while ABA still outperforms ReAct by around 20% on average.

Table 2: Statistics on policy efficiency in ALFWorld

|  | ReAct | ABA |
|---|---|---|
| Length [Succ] | $18.2 \pm 9.7$ | $11.5 \pm 3.3$ |
| Length [All] | $29.3 \pm 17.1$ | $16.9 \pm 13.8$ |
| # Actions | $13.1 \pm 8.3$ | $6.3 \pm 2.6$ |

Besides the final performance, we also include statistics on policy efficiency in Table 2. We logged the trajectories with GPT-3.5 backbone and report the averaged episode length of successful trails and all trails. The episode length includes physical actions, thinking steps and asking steps. We also report the number of physical actions taken to interact with the environment. It can be concluded that, even with additional action type (i.e., asking), ABA still enjoys a shorter trajectory. What's more, ABA can finish the task by taking more than 50% less actions. This proves that ABA not only have better final performance, but also enjoy better efficiency. We show example trajectories and qualitative analysis in AppendixB. Since we use human models to reduce human involvement, we also test the accuracy of the human model. For more details, refer to AppendixC.

To further assess the capabilities of ABA, we carefully design two variants of ALFWorld, namely Ambiguous ALFWorld and Multiround ALFWorld. These modifications pose new challenges. For Ambiguous ALFWorld, we adjust the task descriptions and reward functions to introduce ambiguity, which deliberately left some aspects open-ended to necessitate the agent gathering additional information. For Multiround ALFWorld, we assign a new task for the agent within the same room once the previous task has completed, which allows us to test whether the agent can remember previously known information and avoid asking repeatedly. For more details about the two environments, please refer to D. In this section, we compare ABA and ABA-FT with ReAct (Yao et al., 2022). For ABA and ReAct, we utilize human-annotated examples, while for ABA-FT, we manually design an expert policy to collect data. Additional details can be found in Appendix E.

The results are shown in Figure 2. Note that two environments correspond to different $y$-axis. Though the two variants are more complex than original ALFWorld, we find that both ABA and ABA-FT consistently exhibit superior performance, while ReAct fails to complete the task in most scenarios. This again highlights the significance of actively questioning for necessary information, and proves that taking consistent actions to gather necessary information in various complex environments is challenging for current agent. For both of the environments, the ABA-FT outperforms ABA, and the margin is significantly larger in Multiround ALFWorld. This observation aligns with our analysis in Section 3.2.2 that ABA-FT may further improve performance especially when trajectories are long. It is also worth noticing that, for ABA-FT, we only use 1500 and 500 trajectories to finetune one model for all the tasks for each environment. When compared with classical Imitation Learning method like BUTLER, this only accounts for 1.5% and 0.5% of one dataset (note

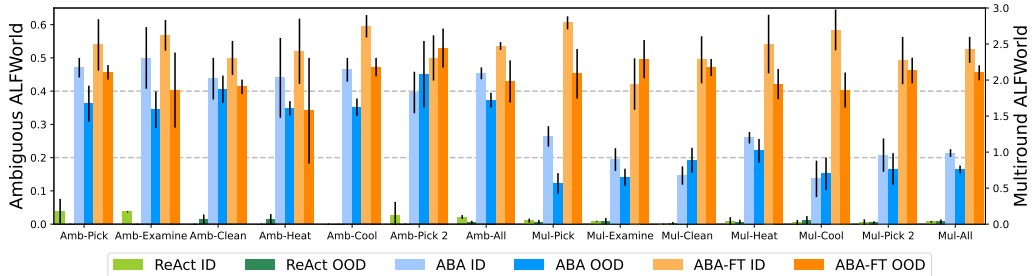

Figure 2: Performance on Ambiguous ALFWorld and Multiround ALFWorld. The prefixes of the $x$-label `Amb-` and `Mul-` refer to Ambigous ALFWorld and multi-round ALFWorld, and two environments correspond to different $y$-axis. The legend suffixes `-ID` and `-OOD` refer to performance on in-distribution and out-of-distribution evaluation sets.

BUTLER needs to independently finetune for each task). This proves that even with ABA-FT, we can effectively adapt to new scenarios at a relatively lower cost. Qualitative analysis can be found in Appendix F and Appendix G. We provide example trajectories on Ambiguous ALFWorld and Multiround ALFWorld to demonstrate that, both ABA and ABA-FT are able to ask pertinent questions to gather necessary information, while ReAct struggles to conduct consistent exploration. What's more, from Appendix G, we show that the agent is capable of recalling previously acquired information and leveraging it in the following tasks.

## 4.2 ROBOT ARM MANIPULATION TASKS

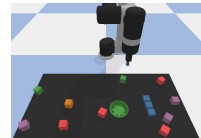

| | Task 1 | | | Task 2 | | | Task 3 |
|---|---|---|---|---|---|---|---|
| | $x = 3$ | $x = 4$ | $x = 5$ | $y = 2$ | $y = 3$ | $y = 4$ | $x = 4, y = 3$ |
| ReAct | 65 | 70 | 55 | 95 | 35 | 5 | 30 |
| ABA | **85** | **85** | **90** | **100** | **95** | **90** | **90** |

Figure 3: *Left*: Visualization of an example task instance in Task 3. *Right*: Average success rates across 20 trails on three manipulation tasks. Here, $x$ and $y$ refers to parameters in each task.

In this section, we further explore whether ABA is able to handle control tasks. Instead of text-based observations and actions, we design the agent to operate on numerical level by directly outputs coordinates of control commands. We consider it poses unique challenges since the asking and answering are formulated in natural language, which is relatively abstract with respect to controlling. Therefore, after proposing the questions, the agent needs to reason from answers to get the detailed information to guide decision-making (e.g., from the answer "the second cube from the left", the agent should be able to reason which cube to move, and output corresponding numerical commands).

We design three different tasks with unique challenges (see Figure 3 for visualization of an example task instance): (1) **Task 1**: the agent is instructed to move the red block into the green bowl. However, there might be $x$ such red blocks, but only one of them is the goal object and the task is considered success when only the correct block is in the bowl. Furthermore, we restricted the human model to answer in realtive position (e.g., "the second red block from the left") when asked. Therefore, the agent needs to reason to get the information according to the answer. (2) **Task 2**: the agent is instructed to place some of the blocks on $y$ corresponding bases. The bases refer to the fixed blue blocks on the table, and each base block has a different corresponding color. The task is considered success when all the bases have a corresponding color block placed on it. We make the agent to only ask one piece of information at a time (e.g., it can only as for one color correspondance) and restrict the total amount of questions to be $y - 1$. Therefore, the agent must inferred the last color correspondance without asking to finish the task. (3) **Task 3**: the agent is instructed to finish the two tasks above and query diverse information. For experiment details, please refer to Appendix J.

The results can be found in Figure 3. We compare our method with ReAct, and use GPT-4 API as backbones for both of the method. For each trail, the placement of objects are randomly sampled. We report the average success rate across 20 trails for each setting. The results demonstrate that ABA outperforms ReAct in all the settings. In Task 1, we show ABA is able to reason the control actions from the relative position language, while ReAct needs to try the block one-by-one (see Appendix J for example trajectories). In Task 2 and Task 3, the performance gap is obvious especially when $y$ is relatively large. As $y$ increases, ABA performances stay relatively stable and remain above 90%, while the baseline performance gradually deteriorates (from 95% when $y = 2$ to 5% when $y = 4$). This proves that, under large exploration space, even current state-of-the-art LLMs like

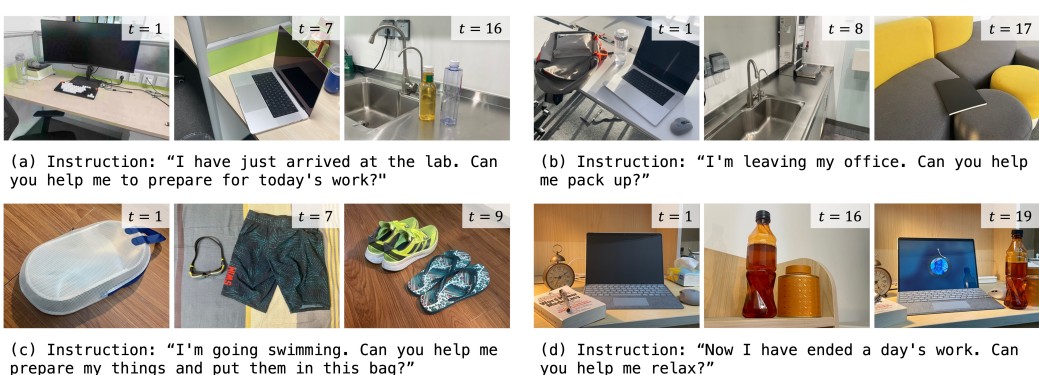

(a) Instruction: "I have just arrived at the lab. Can you help me to prepare for today's work?"

(b) Instruction: "I'm leaving my office. Can you help me pack up?"

(c) Instruction: "I'm going swimming. Can you help me prepare my things and put them in this bag?"

(d) Instruction: "Now I have ended a day's work. Can you help me relax?"

Figure 4: Visualization of real-world open domain embodied tasks. We show four trajectories and their corresponding instructions.

GPT-4 struggle to take consistently explore and gather the information. These findings provide further evidence for the effectiveness of ABA.

### 4.3 REAL-WORLD OPEN DOMAIN EMBODIED TASKS WITH IMAGE INPUTS

In this section, we carry out qualitative experiments to evaluate ABA in real-world scenarios with image inputs and human involvement to examine its ability to handle open-domain embodied decision-making tasks. We present the visualization in Figure 4. Real-world scenarios pose several challenges: First, real-world tasks and scenarios are more flexible and complex than a simulator, which requires up to 19 steps and 7 question asking to solve in our case. The diversity and complexity force us to test ABA in a zero-shot fashion since it is almost impossible to collect examples in advance. Second, the involvement of humans allows open domain question asking compared with human models, and it creates a more realistic scenario mimicking deploying ABA in practice.

However, the lack of a versatile embodied robot makes it hard to autonomously follow the plan proposed by the algorithm. Therefore, we ablate the plan proposal and execution by asking a human volunteer to strictly follow the proposed plan and interact with the world. In this way, we can solely focus on testing whether ABA can propose grounded questions and feasible plans to finish the task.

We build four real-world scenarios including a laboratory, an office, a bedroom, and a dormitory for evaluation. As for implementation, we use InstructBLIP (Liu et al., 2023) with Vicuna-13B Chiang et al. (2023) as the caption model, and use GPT-4 API as our decision-making backbone. We use $K = 1$ human labeled trajectory as in-context examples (Figure 4 (a)) and test ABA on three different scenarios (Figure 4 (b-d)).

ABA successfully finished all three scenarios. We present the keyframes and instructions in Figure 4 and the full trajectories in Appendix K. The results qualitatively demonstrate that ABA is able to propose open-domain grounded questions and feasible plans. To be more specific, here are some excerpted questions. For (b), $t = 1$: "Besides the laptop, do you need to pack anything else?" and for (d), $t = 1$: "What kind of activities do you prefer to relax? Do you want to read the book, listen to music, or do something else?", the agent is able to propose open-domain questions by making suggestions while gathering the preference. For (c), $t = 9$: "Which pair of flip-flops would you like me to take, the green and yellow ones or the blue and white ones?" the agent can ask questions according to the image observations. Though the scenarios are rather limited, we believe the trajectories can showcase the potential practical value of ABA and augurs a promising direction for future research.

## 5 RELATED WORKS

### 5.1 LANGUAGE AGENT

Natural language modeling pre-trained on large-scale unstructured text corpus has seen tremendous success in a variety of applications, including downstream NLP tasks (Radford et al., 2019; Devlin et al., 2018; Brown et al., 2020), logic reasoning (Zhao et al., 2023; Cobbe et al., 2021; Shen et al., 2021), and human-AI coordination (Bubeck et al., 2023; Hu & Sadigh, 2023). The rich information contained in LLMs as an implicit knowledge base also catalyzes the research on in-context learning (Shin et al., 2022; Xie et al., 2021) and prompting (Brown et al., 2020; Wei et al., 2022) that prepend instructions and a few examples to the input of LLMs. However, the time and memory complexity for encoding the prompt is quadratic in the length of the interaction history, such as all the previ-

ous trajectories in embodied decision-making, which can increase the burden of the self-attention mechanism and even exceed the token limitations of LLMs. Despite the techniques introduced to address this issue (Mu et al., 2023; Bulatov et al., 2023), the proposed ABA-IL is inspired by the recent studies on fine-tuning LLMs (Houlsby et al., 2019; Hu & Sadigh, 2023; Lialin et al., 2023), especially those that leverage decision-making signals to train language agents that satisfy certain goals (Carta et al., 2023; Snell et al., 2022a;b). LLMs have also shown great potential for task planning (Huang et al., 2022b; Lin et al., 2023; Huang et al., 2022a; Wang et al., 2023; Li et al., 2022; Singh et al., 2022a; Carta et al., 2023). However, recent criticisms are made on the planning abilities of LLMs (Bubeck et al., 2023; Valmeekam et al., 2022; 2023). They show that LLMs can get stuck in long-horizon decision-making tasks and the resulting search procedure often degrades to exhaustive search over the large state and action spaces. While pure LLM planning remains a highly challenging open problem, in this work, we investigate the capacity of LLM agents to actively gather information with humans in the loop.

## 5.2 Embodied decision-making with Human-in-the-Loop

Some existing works have also studied the scenarios with human-in-the-loop. They query humans for extra information to guide decision-making. A majority of works (Nguyen & Daumé III, 2019; Nguyen et al., 2019; Singh et al., 2022b; Da Silva et al., 2020) directly ask humans for oracle actions, and most of the works focus on visual navigation (Gao et al., 2021; 2023). Unlike recent works in visual navigation (Zhao et al., 2022; Chen et al., 2022), they either learn when to ask for oracle actions (Da Silva et al., 2020; Singh et al., 2022b; Chi et al., 2020), query where the goal is in the view Zhang et al. (2023), or learn to leverage language instructions (Nguyen & Daumé III, 2019; Nguyen et al., 2019). Nguyen et al. (2022) asks for new descriptions of current states and (sub-)goals in a POMDP. For asking and answering in language, Huang et al. (2022b) engages humans in active scene description, allowing the LLM to consider human feedback of scene, task, and preferences as inputs. Liu et al. (2022) asks 3-word-templated selected by an RL agent and calculate bi-gram similarity. Yao et al. (2022) also mentions the engagement of humans. Still, it requires humans to supervise and modify the model's output in real-time , which is different from other HTIL settings.

Existing works include human-in-the-loop of decision-making either (1) directly asking for numerical vectors such as actions/states (Da Silva et al., 2020; Singh et al., 2022b; Nguyen et al., 2022), or (2) inquiring humans to give exhaustive instruction and learn to convert them to actions (Nguyen & Daumé III, 2019; Nguyen et al., 2019). Different from these works, we only put a minimal burden on humans and ask them for natural language information rather than detailed action instructions. Instead of considering human feedback as the scene (or task, preference) descriptor in the decision-making pipeline (Huang et al., 2022b), we formulate the setting as *Contextual MDP with Human / External Information Sources in the Loop*, which elaborates the effects of asking via context $c$ and allow the agent to query a broader range of information to gather information. Besides, unlike Liu et al. (2022), we focus on zero-shot adaptation settings and propose more natural end-to-end methods to circumvent the needs of template and similarity designing. Concurrently, Ren et al. (2023) proposes to use an explicit uncertainty module that helps decide when to trigger asking. However, ABA allows for seamlessly integrate asking and acting in an end-to-end manner, and ABA-FT allows better data scalbility.

## 6 Conclusion and Discussion

In this paper, we focus on the setting where the agent can actively query for additional pertinent information from external sources using natural language while interacting in the environments. To formalize this problem, we propose *Contextual MDP with Human / External Information Sources in the Loop*. Then, we propose *Asking Before Acting* (ABA), a method that empowers the agent to ask various questions to gather diverse information. We conduct extensive experiments involving a spectrum of environments including text-based household everyday tasks, robot arm manipulation tasks, and real world open domain image based embodied tasks on various models. The results demonstrate that, ABA exhibits substantial advantages on both performance and efficiency over baseline LLM agents. Further finetuning ABA with reformulated metadata (ABA-FT) promotes learning the asking rationale and allows for additional enhancements. We believe our work serves as one of the first papers in combining LLMs (VLMs) with human in the loop decision making, and we hope ABA can be inspiring. However, in real world experiments, we currently use a perfect controller. In the future, we believe it is important to further study how to combine our method with real world robot to conduct more tasks.

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

## A  DESIGN AND DETAILS ABOUT HUMAN MODEL

In this section, we describe the design and details about the human model (or other information sources) $\mathcal{H}$. To minimize human involvement, during the evaluation phase, we implement $\mathcal{H}$ via another language model which is instructed to respond to questions based on the provided information. To incorporate prior knowledge about the current room, we extract the information about the object placement from the simulator and transform it into a descriptive paragraph. This paragraph is then fed to $\mathcal{H}$. Specifically, we use Vicuna-7B (Chiang et al., 2023) to implement $\mathcal{H}$. Using a pretrained LLM as $\mathcal{H}$ allows for answering questions in free-form language based on the information provided, which acts just like humans.

To better demonstrate, we provide an example for the ALFWorld experiment in Section 4.1. In ALFWorld, the context $c$ mainly refers to the initial mappings of the object placement. For different rooms, the initial mappings are, therefore, different. We slightly abuse the notations about $c$ here since the agent may replace the objects. Under this mapping, we can directly get the ground truth object locations from the simulator, which are unobservable to the agent. Then, we use a rule-based conversion to convert that list to a string of "$A$ is in $B$", where $A$ refers to the object, while $B$ refers to the place containing the object.

Here is an example. After converting, we derive a descriptive paragraph like:

```
        bowl 2 is in diningtable 2.  saltshaker 2 is in
    sidetable 1.  spatula 1 is in countertop 1.  pot
    1 is in stoveburner 4.  spatula 2 is in drawer 1.
    dishsponge 3 is in diningtable 2.  peppershaker 1 is
    in cabinet 2.  tomato 4 is in sidetable 1.  knife 1
    is in diningtable 3.  cup 1 is in sidetable 1.  bread
    2 is in diningtable 3.  spatula 3 is in diningtable
    2.  pan 1 is in cabinet 4.  tomato 3 is in fridge
    1.  potato 1 is in sinkbasin 1.  peppershaker 3 is
    in diningtable 3.  apple 1 is in fridge 1.  saltshaker
    1 is in cabinet 4.  fork 2 is in drawer 1.  spoon 1 is
    in sidetable 1.  egg 1 is in fridge 1.  lettuce 1 is
    in sidetable 1.  plate 1 is in diningtable 2.
```

Whenever the agent poses a question, $\mathcal{H}$ is tasked with providing an answer based on this paragraph. For instance, the agent may learn to ask:

```
    Where can I find the dishsponge?
```

Then, in this example, the input to $\mathcal{H}$ will be (1) an instruction that tells the model to provide the answers (in gray); (2) a descriptive paragraph (in black); (3) the question proposed by the agent (in blue).

```
        Read the following paragraph and answer questions:
    bowl 2 is in diningtable 2.  saltshaker 2 is in
    sidetable 1.  spatula 1 is in countertop 1.  pot
    1 is in stoveburner 4.  spatula 2 is in drawer 1.
    dishsponge 3 is in diningtable 2.  peppershaker 1 is
    in cabinet 2.  tomato 4 is in sidetable 1.  knife 1
    is in diningtable 3.  cup 1 is in sidetable 1.  bread
    2 is in diningtable 3.  spatula 3 is in diningtable
    2.  pan 1 is in cabinet 4.  tomato 3 is in fridge
    1.  potato 1 is in sinkbasin 1.  peppershaker 3 is
    in diningtable 3.  apple 1 is in fridge 1.  saltshaker
    1 is in cabinet 4.  fork 2 is in drawer 1.  spoon 1 is
    in sidetable 1.  egg 1 is in fridge 1.  lettuce 1 is
    in sidetable 1.  plate 1 is in diningtable 2.     The
    questions is:  Where can I find the dishsponge?
```

Then, the pretrained LLM (e.g., Vicuna-7B Chiang et al. (2023) in our case), $\mathcal{H}$ will provide the answers since it can follow the instructions. In our case, the answer is:

```
    dishsponge 1 is in garbagecan 1, dishsponge 2 is in
drawer 3, dishsponge 3 is in drawer 4.
```

To enhance the performance and improve the accuracy, we provide a few in-context examples and use the prompting format as in Vicuna Chiang et al. (2023). Please refer to Appendix I for details.

It's worth noting that, though we use a language model to simulate humans in our evaluation phase, this design allows for the straightforward replacement of the current language model with human or alternative information sources for more appropriate answers. The performance of our method will be further improved if $\mathcal{H}$ can provide more accurate and meaningful answers.

## B  DETAILS, EXAMPLES AND QUALITATIVE ANALYSIS FOR ALFWORLD EXPERIMENTS

For ALFWorld experiment, now we describe the benchmark and baselines in details.

ALFWorld (Shridhar et al., 2021) is an interactive text-based environment with aligned embodied worlds, which contains six types of different everyday household tasks from ALFRED Shridhar et al. (2020) encompassing activities such as picking and placing, examining in light, cleaning, etc. In each episode, the agent is deployed in a new room and assigned specific tasks to accomplish. All observations and actions are in natural language.

As for baselines, we select two representative baselines, BUTLER (Shridhar et al., 2021) and ReAct (Yao et al., 2022). BUTLER (Shridhar et al., 2021) is an imitation learning based method without LLM. It trains an independent model for each task, while each of the model uses a substantial dataset of $10^5$ expert trajectories. ReAct (Yao et al., 2022) is an LLM-based method synergizing reasoning and acting to take actions. It is worth noticing that, for ReAct(Yao et al., 2022) and ABA (we may also use the term ABA interchangeably), we use the same general pretrained LLM across all tasks with $K = 2$ in-context examples. As for the implementation, we try Vicuna-7B Chiang et al. (2023) and GPT-3.5 OpenAI (2023) as backbones for both ReAct and our method, and we incorporate the reasoning process when making decisions (Yao et al., 2022; Ahn et al., 2022). For a fair comparison, we use the same scoring method to select actions for both our method and ReAct.

This section provides some examples and qualitative analysis for ALFWorld experiments in Section 4.1.

First, we provide an episode trajectory of our method (ABA-IC), which is randomly sampled from the evaluation set:

```
    Obs 1:     You are in the middle of a room.  Looking
quickly around you, you see a bed 1, a diningtable
1, a drawer 4, a drawer 3, a drawer 2, a drawer 1, a
garbagecan 1, a sidetable 2, and a sidetable 1.  Your
task is to:  put a mug in sidetable.
    Act 1: think:  To solve the task, I need to find and
take a mug, then put it in sidetable.  But where is
the mug?  Let me ask that person.
    Obs 2: OK.
    Act 2: ask:  Where is the mug?
    Obs 3:         mug 1 is in diningtable 1, mug 3 is in
diningtable 1, mug 2 is in diningtable 1.
    Act 3:   think:  We can go to diningtable 1 and take
the mug 1, then put it in sidetable.
    Obs 4: OK.
    Act 4: go to diningtable 1
    Obs 5: On the diningtable 1, you see a creditcard 3,
a creditcard 2, a keychain 3, a keychain 2, a mug 3,
a mug 2, a mug 1, a pen 2, a pen 1, a pencil 3, and a
pencil 1.
```

```
    Act 5:  think:  Now I find the mug 1.  Next, I need
to take it, then put it in sidetable.
    Obs 6: Ok.
    Act 6: take mug 1 from diningtable 1
    Obs 7: You pick up the mug 1 from the diningtable 1.
    Act 7:  think:  Now I take a mug 1.  Next, I need to
put it in sidetable.
    Obs 8: Ok.
    Act 8: go to sidetable 1
    Obs 9: On the sidetable 1, you see a keychain 1.
    Act 9: put mug 1 in/on sidetable 1
    Obs 10: You put the mug 1 in/on the sidetable 1.
```

It can be concluded that, for our method, the agent can identify the correct context that is related to the task (i.e., the location of the mug) and then summarize the question and ask a pertinent question (i.e., "Where is the mug?"). After that, the agent can integrate the information in the answer (i.e., notice that mug 1 is in diningtable 1) and act according to that information (i.e., go to diningtable 1, take the mug, and place it in the specific location).

For comparison, we provide the trajectory of the same episode of ReAct (Yao et al., 2022):

```
    Obs 1:     You are in the middle of a room.  Looking
quickly around you, you see a bed 1, a diningtable
1, a drawer 4, a drawer 3, a drawer 2, a drawer 1, a
garbagecan 1, a sidetable 2, and a sidetable 1.  Your
task is to:  put a mug in sidetable.
    Act 1: think:  To solve the task, I need to find and
take a mug, then put it in sidetable.
    Obs 2: OK.
    Act 2:    think:  First I need to find a mug.  A mug
is more likely to appear in drawer (1-4), sidetable
(1-2).  I can check one by one, starting with drawer
1.
    Obs 3: OK.
    Act 3: go to drawer 1
    Obs 4: The drawer 1 is closed.
    Act 4: open drawer 1
    Obs 5: You open the drawer 1.  The drawer 1 is open.
In it, you see a cellphone 1.
    Act 5:  think:  Now I find a mug (1).  Next, I need
to take it.
    Obs 6: OK.
    Act 6: go to drawer 2
    Obs 7: The drawer 2 is closed.
    Act 7: open drawer 2
    Obs 8: You open the drawer 2.  The drawer 2 is open.
In it, you see a creditcard 1.
    Act 8:  think:  Now I find a mug (2).  Next, I need
to take it.
    Obs 9: OK.
    Act 9: go to drawer 3
    Obs 10: On the drawer 3, you see a cellphone 2.
    Act 10:  think:  Now I find a mug (3).  Next, I need
to take it.
    Obs 11: OK.
    Act 11: go to drawer 4
    Obs 12: On the drawer 4, you see nothing.
    Act 12:  think:  Now I find a mug (4).  Next, I need
to take it.
```

*Obs 13:* OK.
*Act 13:* go to sidetable 1
*Obs 14:* On the sidetable 1, you see a keychain 1.
*Act 14:* think: Now I find a mug (1). Next, I need to put it in sidetable 1.
*Obs 15:* OK.
*Act 15:* go to sidetable 2
*Obs 16:* On the sidetable 2, you see a alarmclock 1, a cd 1, and a pencil 2.
*Act 16:* think: Now I find a mug (2). Next, I need to put it in sidetable 2.
*Obs 17:* OK.
*Act 17:* go to drawer 1
*Obs 18:* The drawer 1 is open. In it, you see a cellphone 1.
*Act 18:* think: Now I find a mug (1). Next, I need to put it in drawer 1.
*Obs 19:* OK.
*Act 19:* go to drawer 2
*Obs 20:* The drawer 2 is open. In it, you see a creditcard 1.
*Act 20:* think: Now I find a mug (2). Next, I need to put it in drawer 2.
*Obs 21:* OK.
*Act 21:* go to drawer 3
*Obs 22:* On the drawer 3, you see a cellphone 2.
*Act 22:* think: Now I find a mug (3). Next, I need to put it in drawer 3.
*Obs 23:* OK.
*Act 23:* go to drawer 4
*Obs 24:* On the drawer 4, you see nothing.
*Act 24:* think: Now I find a mug (4). Next, I need to put it in drawer 4.
*Obs 25:* OK.
*Act 25:* go to bed 1
*Obs 26:* On the bed 1, you see a book 2, a book 1, a laptop 2, a laptop 1, and a pillow 1.
*Act 26:* think: Now I find a mug (1). Next, I need to put it on bed 1.
*Obs 27:* OK.
*Act 27:* go to sidetable 1
*Obs 28:* On the sidetable 1, you see a keychain 1.
*Act 28:* think: Now I find a mug (1). Next, I need to put it on sidetable 1.
*Obs 29:* OK.
*Act 29:* go to sidetable 2
*Obs 30:* On the sidetable 2, you see a alarmclock 1, a cd 1, and a pencil 2.
*Act 30:* think: Now I find a mug (1). Next, I need to put it on sidetable 2.
*Obs 31:* OK.
*Act 31:* go to drawer 1
*Obs 32:* The drawer 1 is open. In it, you see a cellphone 1.
*Act 32:* think: Now I find a mug (1). Next, I need to put it in drawer 1.
*Obs 33:* OK.
*Act 33:* go to drawer 2

*Obs 34:*     The drawer 2 is open.  In it, you see a creditcard 1.
*Act 34:* think:  Now I find a mug (1).  Next, I need to put it in drawer 2.
*Obs 35:* OK.
*Act 35:* go to drawer 3
*Obs 36:* On the drawer 3, you see a cellphone 2.
*Act 36:* think:  Now I find a mug (2).  Next, I need to put it in drawer 3.
*Obs 37:* OK.
*Act 37:* go to drawer 4
*Obs 38:* On the drawer 4, you see nothing.
*Act 38:* think:  Now I find a mug (3).  Next, I need to put it in drawer 4.
*Obs 39:* OK.
*Act 39:* go to bed 1
*Obs 40:*  On the bed 1, you see a book 2, a book 1, a laptop 2, a laptop 1, and a pillow 1.
*Act 40:* think:  Now I find a mug (1).  Next, I need to put it on bed 1.
*Obs 41:* OK.
*Act 41:* go to sidetable 1
*Obs 42:* On the sidetable 1, you see a keychain 1.
*Act 42:* think:  Now I find a mug (1).  Next, I need to put it on sidetable 1.
*Obs 43:* OK.
*Act 43:* go to sidetable 2
*Obs 44:*  On the sidetable 2, you see a alarmclock 1, a cd 1, and a pencil 2.
*Act 44:* think:  Now I find a mug (1).  Next, I need to put it on sidetable 2.
*Obs 45:* OK.
*Act 45:* go to drawer 1
*Obs 46:*     The drawer 1 is open.  In it, you see a cellphone 1.
*Act 46:* think:  Now I find a mug (1).  Next, I need to put it in drawer 1.
*Obs 47:* OK.
*Act 47:* go to drawer 2
*Obs 48:*     The drawer 2 is open.  In it, you see a creditcard 1.
*Act 48:* think:  Now I find a mug (1).  Next, I need to put it in drawer 2.
*Obs 49:* OK.
*Act 49:* go to drawer 3
*Obs 50:* On the drawer 3, you see a cellphone 2.
*Act 50:* think:  Now I find a mug (2).  Next, I need to put it in drawer 3.
*Obs 51:* OK.

It can be concluded that the agent failed to provide the right plan at first (i.e., Action 1,2), since none of these locations has a mug. Moreover, even with the plans, it failed to yield coherent and correct actions in later steps. For instance, in Action 5, 8, 10, 12, etc., the agent is constantly confused by hallucination (it believes that "Now I find a mug" while there wasn't a mug actually). It also got stuck in loops (it repeatedly visited drawer 1,2,3,4, sidetable 1,2, and bed 1). These together prevent the agent from finishing the task.

To further understand the experiment results, we provide more examples for our method and ReAct.

For our method:

    *Obs 1:*    You are in the middle of a room. Looking quickly around you, you see a bathtubbasin 1, a countertop 1, a drawer 4, a drawer 3, a drawer 2, a drawer 1, a dresser 1, a garbagecan 1, a handtowelholder 2, a handtowelholder 1, a shelf 2, a shelf 1, a sinkbasin 1, a toilet 1, a toiletpaperhanger 1, and a towelholder 1. Your task is to: put some spraybottle on toilet.

    *Act 1:* think: To solve the task, I need to find and take a spraybottle, then put it on the toilet. But where is the spraybottle? Let me ask that person.

    *Obs 2:* OK.

    *Act 2:* ask: Where is the spraybottle?

    *Obs 3:* spraybottle 3 is in countertop 1, spraybottle 4 is in dresser 1, spraybottle 2 is in shelf 1.

    *Act 3:*   think: We can go to countertop 1 and take the spraybottle 3, then put it on the toilet.

    *Obs 4:* OK.

    *Act 4:* go to countertop 1

    *Obs 5:*   On the countertop 1, you see a mirror 1, a soapbar 1, and a spraybottle 1.

    *Act 5:* think: Now I find the spraybottle 1. Next, I need to take it, then put it on the toilet.

    *Obs 6:* OK.

    *Act 6:* take spraybottle 1 from countertop 1

    *Obs 7:*    You pick up the spraybottle 1 from the countertop 1.

    *Act 7:* think: Now I take a spraybottle 1. Next, I need to put it on the toilet.

    *Obs 8:* OK.

    *Act 8:* go to toilet 1

    *Obs 9:*    On the toilet 1, you see a candle 1, a toiletpaper 2, and a toiletpaper 1.

    *Act 9:*   think: Now I put the spraybottle 1 on the toilet.

    *Obs 10:* OK.

    *Act 10:* put spraybottle 1 in/on toilet 1

    *Obs 11:*  You put the spraybottle 1 in/on the toilet 1.

For ReAct:

    *Obs 1:*    You are in the middle of a room. Looking quickly around you, you see a bathtubbasin 1, a countertop 1, a drawer 4, a drawer 3, a drawer 2, a drawer 1, a dresser 1, a garbagecan 1, a handtowelholder 2, a handtowelholder 1, a shelf 2, a shelf 1, a sinkbasin 1, a toilet 1, a toiletpaperhanger 1, and a towelholder 1. Your task is to: put some spraybottle on toilet.

    *Act 1:* think: To solve the task, I need to find and take a spraybottle, then put it on toilet.

    *Obs 2:* OK.

    *Act 2:*   think: First I need to find a spraybottle. A spraybottle is more likely to appear in drawer (1-4), dresser (1), sinkbasin (1), toilet (1), and towelholder (1). I can check one by one, starting with drawer 1.

    *Obs 3:* OK.

```
    Act 3: go to drawer 1
    Obs 4: The drawer 1 is closed.
    Act 4: open drawer 1
    Obs 5: You open the drawer 1.  The drawer 1 is open.
In it, you see nothing.
    Act 5:       think:  Now I find a spraybottle (not).
Next, I need to take it.
    Obs 6: OK.
    Act 6: go to drawer 2
    Obs 7: The drawer 2 is closed.
    Act 7: open drawer 2
    Obs 8: You open the drawer 2.  The drawer 2 is open.
In it, you see a candle 3.
    Act 8:       think:  Now I find a spraybottle (not).
Next, I need to take it.
    Obs 9: OK.
    Act 9: go to drawer 3
    Obs 10: The drawer 3 is closed.
    Act 10: open drawer 3
    Obs 11:      You open the drawer 3.  The drawer 3 is
open.  In it, you see nothing.
    Act 11:      think:  Now I find a spraybottle (not).
Next, I need to take it.
    Obs 12: OK.
    Act 12: go to drawer 4
    Obs 13: The drawer 4 is closed.
    Act 13: open drawer 4
    Obs 14:      You open the drawer 4.  The drawer 4 is
open.  In it, you see nothing.
    Act 14:      think:  Now I find a spraybottle (not).
Next, I need to take it.
    Obs 15: OK.
    Act 15: go to dresser 1
    Obs 16:      On the dresser 1, you see a candle 2, a
soapbottle 3, a soapbottle 2, and a spraybottle 4.
    Act 16: take spraybottle 4 from dresser 1
    Obs 17:      You pick up the spraybottle 4 from the
dresser 1.
    Act 17: think:  Now I find a spraybottle (4).  Next,
I need to put it in/on toilet 1.
    Obs 18: OK.
    Act 18: go to toilet 1
    Obs 19:       On the toilet 1, you see a candle 1, a
toiletpaper 2, and a toiletpaper 1.
    Act 19: put spraybottle 4 in/on toilet 1
    Obs 20:  You put the spraybottle 4 in/on the toilet
1.
```

From the above examples, our method can ask proper questions and act accordingly. Though $\mathcal{H}$ provides information with slight error (e.g., it mentions that spraybottle 3 is in countertop 1, but only spraybottle 1 is in countertop 1), the agent is robust to such error and successfully adjust its behavior after observing the objects on countertop 1 (i.e., action 5, 6, 7).

As for ReAct, it successfully visited four drawers and finally found the spraybottle at dresser 1. However, first, it failed to list every possible container for the spraybottle (i.e., action 2, it omits countertop, shelf, etc.). In the reasoning step, we observe an interesting pattern (i.e., in Action 5, 8, 11, 14): "Now I find a spraybottle (not). Next, I need to take it", which seems inconsistent (though it does not affect the next step). Moreover, though the agent finally finds the spraybottle and

completes the task successfully, it is inefficient and slow to search every possible location: ReAct takes 20 steps. In comparison, our method only takes 10 steps to finish the task.

Four above examples demonstrate that, first, it is challenging to learn a information-gathering policy especially in unfamiliar environments, due to the complexity of the environment. Moreover, even if the agent manage to follow this policy, the information-gathering phase can be inefficient, which needs to exhaustively search every possible position. In contrast, our method succeeds in proposing proper questions and then acting accordingly, which improve the success rate as well as the efficiency. This proves our method's efficacy.

## C  Accuracy of Human Model

In this section, we test the accuracy of the human model. We test across 8 randomly sampled scenarios in ALFWorld with 5 questions each, and see whether the human model can answer correctly.

The correct rates as shown in the following table. The results of models implemented in Vicuna 7B:

| | Accuracy | | | | | | | | All |
|---|---|---|---|---|---|---|---|---|---|
| Human Model | 80 | 80 | 80 | 80 | 60 | 60 | 80 | 60 | $72.5 \pm 9.7$ |

## D  Details about the two ALFWorld variants: Ambiguous ALFWorld and Multiround ALFWorld

In Ambiguous ALFWorld, we manually adjusted the task descriptions and reward functions to introduce ambiguity. Instead of providing precise task descriptions, we deliberately left some aspects open-ended, thereby necessitating the agent to gather additional information for successful completion. For instance, in ALFWorld, the task "put a mug on the shelf" is typically considered accomplished as long as any mug is placed on the shelf (there might be multiple mugs in the room). But in this modified setting, the task is only deemed completed when a specific mug is put on the shelf. To complete this task, one can either enumerate all possibilities accordingly until the correct one is identified or directly ask for further clarification about the task.

To further test whether the agent is able to remember the previously known information and avoid asking repeatedly, we introduce multiround ALFWorld. In previous experiments, the episode ends as long as the current task is completed. subsequently, in the next episode, the environment will reset to another room with a different layout. In Multiround ALFWorld, after one task is completed, we randomly sample a new task for the agent to undertake within the same room for multiple rounds. This adjustment enables the agent to familiarize itself with the object placement and provides an opportunity to test its capability to remember and refrain from repetitive questioning. For instance, suppose the agent has previously visited the sidetable to complete a previous task and happened to see there is a mug, or the agent has previously ask about the location of the mug, when the agent is tasked to bring a mug, it can directly go to the location without the need for further inquiries. In this environment, instead of measuring the success rate as in previous experiments, we assign a reward $r = 1$ upon the completion of each task and measure the total reward after $T$ steps.

## E  Details about the Data Collection

In this section, we provide details about how the data is collected and training as mentioned of two ALFWorld variants described in Section 4.1.

As for in-context examples used in ABA-IC, we manually interact with the environment and try to finish the tasks. We ask questions related to the tasks, and answer the questions ourselves by checking the ground truth states in the simulator. Beside the questions, we also add reasoning steps as in Yao et al. (2022) and select actions according to the information we have. Once completing the task, we take down all the actions and observations and use them as in-context examples.

As for ABA-FT, we design a rule-based policy according to the PDDL planning trajectories provided along with the environment. Specifically, we integrate the PDDL trajectories and the ground truth states within the simulator to find out what we should do to finish the tasks. Then, we extract the ground truth placements of the necessary objects from the simulator, and we write template-based

questions to query this information and provide corresponding answers as observations. We also write chain-of-thought reasoning steps. As mentioned in Section 3.2.2, we manually inject noises by randomly inserting noisy actions at probability $p = 0.2$. These noisy actions are randomly sampled from the action space. The planning trajectories are also modified accordingly to correctly finish the task. Finally, we organize these questions, reasoning, and interactive actions to get a list of actions. When the actions belong to asking for reasoning, we use provided answers or simply "OK." as observations. When the actions aim to interact with the environment, we use environmental feedback as the observations. As for Ambiguous ALFWorld, we use $K = 2$ for ABA-IC and collect 1500 trajectories for ABA-FT. As for Multiround ALFWorld, we use $K = 1$ for ABA-IC and collect 500 trajectories for ABA-FT. As for training, we use LoRA (Hu et al., 2021) with $r = 16$ and a learning rate of $1e - 5$. We train the model with 3000 steps for Ambiguous ALFWorld and 6000 steps for Multiround ALFWorld.

## F  EXAMPLES AND QUALITATIVE ANALYSIS FOR ALFWORLD WITH AMBIGUOUS TASKS

This section provides some examples and qualitative analysis for ALFWorld with Ambiguous Tasks experiments in Section 4.1.

First, we provide an episode trajectory of ReAct, which is randomly sampled from the evaluation set:

> *Obs 1:*  You are in the middle of a room.  Looking quickly around you, you see a cabinet 10, a cabinet 9, a cabinet 8, a cabinet 7, a cabinet 6, a cabinet 5, a cabinet 4, a cabinet 3, a cabinet 2, a cabinet 1, a coffeemachine 1, a countertop 2, a countertop 1, a diningtable 2, a diningtable 1, a drawer 3, a drawer 2, a drawer 1, a fridge 1, a garbagecan 1, a microwave 1, a sinkbasin 1, a stoveburner 4, a stoveburner 3, a stoveburner 2, a stoveburner 1, and a toaster 1.  Your task is to:  put a hot egg in diningtable.
> *Act 1:*  think:  To finish the task, I need to find an egg, then heat it with microwave, then put it in/on diningtable.
> *Obs 2:* OK.
> *Act 2:* think:  First I need to find an egg.  An egg is more likely to appear in stoveburner 2, countertop 1-3, diningtable 1, and cabinet 9-10.  But I have checked stoveburner 2 and countertop 1-3.  I can check the remaining places one by one, starting with cabinet 9.
> *Obs 3:* OK.
> *Act 3:* go to cabinet 9
> *Obs 4:* The cabinet 9 is closed.
> *Act 4:* think:  An egg does not appear in cabinet 9.
> *Obs 5:* OK.
> *Act 5:* go to cabinet 10
> *Obs 6:* The cabinet 10 is closed.
> *Act 6:* think:  An egg does not appear in cabinet 10.
> *Obs 7:* OK.
> *Act 7:* go to diningtable 1
> *Obs 8:*  On the diningtable 1, you see a bread 2, a bread 1, a butterknife 2, a dishsponge 1, a fork 1, a potato 1, and a spatula 1.
> *Act 8:*  think:  I have checked all the remaining places, and I have not found an egg.  The task is not finished.
> *Obs 9:* OK.

*Act 9:* think:  I have not found an egg.  This is not a failure, just an unexpected result.
  *Obs 10:* OK.
  *Act 10:*  think:  What should I do now?  The task is not finished.
  *Obs 11:* OK.
  *Act 11:* think:  I could try to find an egg again.
  *Obs 12:* OK.
  *Act 12:* go to stoveburner 1
  *Obs 13:* On the stoveburner 1, you see nothing.
  *Act 13:* go to countertop 1
  *Obs 14:*  On the countertop 1, you see a butterknife 1, and a mug 1.
  *Act 14:* think:  It is possible that an egg is hidden under one of these objects.
  *Obs 15:* OK.
  *Act 15:* go to countertop 2
  *Obs 16:* On the countertop 2, you see a bowl 2, and a knife 1.
  *Act 16:* think:  It is possible that an egg is hidden under one of these objects.
  *Obs 17:* OK.
  *Act 17:* go to countertop 1
  *Obs 18:*  On the countertop 1, you see a butterknife 1, and a mug 1.
  *Act 18:* think:  It is possible that an egg is hidden under one of these objects.
  *Obs 19:* OK.
  *Act 19:* go to countertop 2
  *Obs 20:* On the countertop 2, you see a bowl 2, and a knife 1.
  *Act 20:* think:  It is possible that an egg is hidden under one of these objects.
  *Obs 21:* OK.
  *Act 21:* go to countertop 1
  *Obs 22:*  On the countertop 1, you see a butterknife 1, and a mug 1.
  *Act 22:* think:  It is possible that an egg is hidden under one of these objects.
  *Obs 23:* OK.
  *Act 23:* go to countertop 2
  *Obs 24:* On the countertop 2, you see a bowl 2, and a knife 1.
  *Act 24:* think:  It is possible that an egg is hidden under one of these objects.
  *Obs 25:* OK.
  *Act 25:* go to countertop 1
  *Obs 26:*  On the countertop 1, you see a butterknife 1, and a mug 1.
  *Act 26:* think:  It is possible that an egg is hidden under one of these objects.
  *Obs 27:* OK.
  *Act 27:* go to countertop 2
  *Obs 28:* On the countertop 2, you see a bowl 2, and a knife 1.
  *Act 28:* think:  It is possible that an egg is hidden under one of these objects.
  *Obs 29:* OK.
  *Act 29:* go to countertop 1

*Obs 30:*  On the countertop 1, you see a butterknife
1, and a mug 1.
*Act 30:* think:  It is possible that an egg is hidden
under one of these objects.
*Obs 31:* OK.
*Act 31:* go to countertop 2
*Obs 32:* On the countertop 2, you see a bowl 2, and a
knife 1.
*Act 32:* think:  It is possible that an egg is
*Obs 33:* OK.
*Act 33:* go to countertop 1
*Obs 34:*  On the countertop 1, you see a butterknife
1, and a mug 1.
*Act 34:* think:  It is possible that an egg is hidden
under one of these objects.
*Obs 35:* OK.
*Act 35:* go to countertop 2
*Obs 36:* On the countertop 2, you see a bowl 2, and a
knife 1.
*Act 36:* think:  It is possible that an egg is hidden
under one
*Obs 37:* OK.
*Act 37:* go to countertop 1
*Obs 38:*  On the countertop 1, you see a butterknife
1, and a mug 1.
*Act 38:* think:  It is possible that an egg is hidden
under one of these objects.
*Obs 39:* OK.
*Act 39:* go to countertop 2
*Obs 40:* On the countertop 2, you see a bowl 2, and a
knife 1.
*Act 40:* think:  It is possible that an egg is hidden
under one of these objects.
*Obs 41:* OK.
*Act 41:* go to countertop 1
*Obs 42:*  On the countertop 1, you see a butterknife
1, and a mug 1.
*Act 42:* think:  It is possible that an egg is hidden
under one of these objects.
*Obs 43:* OK.
*Act 43:* go to countertop 2
*Obs 44:* On the countertop 2, you see a bowl 2, and a
knife 1.
*Act 44:* think:  It is possible that an egg is hidden
under one of these objects.
*Obs 45:* OK.
*Act 45:* go to countertop 1
*Obs 46:*  On the countertop 1, you see a butterknife
1, and a mug 1.
*Act 46:* think:  It is possible that an egg is hidden
under one of these objects.
*Obs 47:* OK.
*Act 47:* go to countertop 2
*Obs 48:* On the countertop 2, you see a bowl 2, and a
knife 1.
*Act 48:* think:  It is possible that an egg is hidden
under one of these objects.
*Obs 49:* OK.
*Act 49:* go to countertop 1

```
    Obs 50:    On the countertop 1, you see a butterknife
1, and a mug 1.
    Act 50: think:  It is possible that an egg is hidden
under one of these objects.
    Obs 51: OK.
```

In this episode, the agent failed to find the egg. Initially, it failed to propose correct plans (i.e., action 2). It listed several locations where the eggs could be but not all (e.g., cabinet 1-8, diningtable 2, fridge, etc). Then, it incorrectly thought it had checked some of these places, which suggested there might be hallucinations. These two steps limit the searching range of the agent and prevent the agent from finding the target object. In the following steps, it checked cabinet 9-10 (but without opening the cabinets), stoveburner 1, countertop 1-2, and then got stuck by repeatedly visiting the countertop 1-2.

The episode of the same room setting for our method (ABA-FT) is:

```
    Obs 1:     You are in the middle of a room.  Looking
quickly around you, you see a cabinet 10, a cabinet
9, a cabinet 8, a cabinet 7, a cabinet 6, a cabinet
5, a cabinet 4, a cabinet 3, a cabinet 2, a cabinet 1,
a coffeemachine 1, a countertop 2, a countertop 1, a
diningtable 2, a diningtable 1, a drawer 3, a drawer
2, a drawer 1, a fridge 1, a garbagecan 1, a microwave
1, a sinkbasin 1, a stoveburner 4, a stoveburner 3, a
stoveburner 2, a stoveburner 1, and a toaster 1.  Your
task is to:  put a hot mug in cabinet.
    Act 1: think:  To solve the task, I need to find and
take a mug, then heat it with microwave, then put it
in cabinet.  But where is the mug?  Let me ask that
person.
    Obs 2: OK.
    Act 2: ask:  Where is the mug?
    Obs 3:         mug 1 is in countertop 1, mug 2 is in
microwave 1.
    Act 3:    think:  There are multiple mug.  I need to
ask which mug should be taken.
    Obs 4: OK.
    Act 4: ask:  Which mug do you prefer?
    Obs 5: I mean mug 1.
    Act 5:  think:  Now I understand the task.  I can go
to countertop 1 and take the mug 1, then heat it with
microwave, then put it in cabinet.
    Obs 6: OK.
    Act 6: go to countertop 1
    Obs 7: On the countertop 1, you see a butterknife 1,
and a mug 1.
    Act 7: take mug 1 from countertop 1
    Obs 8: You pick up the mug 1 from the countertop 1.
    Act 8:     think:  Now I take a mug 1.  Next, I need
to go to microwave 1 and heat mug 1, then put it in
cabinet.
    Obs 9: OK.
    Act 9: go to microwave 1
    Obs 10: The microwave 1 is closed.
    Act 10: heat mug 1 with microwave 1
    Obs 11: You heat the mug 1 using the microwave 1.
    Act 11:   think:  Now I heat mug 1.  Next, I need to
put it in cabinet 1.
    Obs 12: OK.
```

```
    Act 12: go to cabinet 1
    Obs 13: On the cabinet 1, you see nothing.
    Act 13: put mug 1 in/on cabinet 1
```

The above trajectory shows that, for our method (ABA-FT), the agent is able to identify the correct object related to the task (i.e., mug) and propose a proper question (i.e., Action 2) to ask the position of the mug, which is also shown in Appendix B. Moreover, when it realized that there were multiple mugs in the room, it further asked for clarification about the task (i.e., Action 4) since in Ambiguous ALFWorld, only the specified object is desirable. This demonstrates that it can ask diverse questions to query various information. Finally, the agent is able to integrate the acquired information into the following decision-making: it knows which object is needed, it can head for the correct mug, heat it, and place it on the target location. In contrast to ReAct trajectory where the agent got stuck before finding the mug, the effiency in qualitative examples further proves our method's efficacy.

As a supplement, we show a typical failure case of our method:

```
    Obs 1:      You are in the middle of a room.  Looking
    quickly around you, you see a cabinet 27, a cabinet
    26, a cabinet 25, a cabinet 24, a cabinet 23, a
    cabinet 22, a cabinet 21, a cabinet 20, a cabinet 19,
    a cabinet 18, a cabinet 17, a cabinet 16, a cabinet
    15, a cabinet 14, a cabinet 13, a cabinet 12, a
    cabinet 11, a cabinet 10, a cabinet 9, a cabinet 8,
    a cabinet 7, a cabinet 6, a cabinet 5, a cabinet 4, a
    cabinet 3, a cabinet 2, a cabinet 1, a coffeemachine
    1, a countertop 2, a countertop 1, a diningtable 1,
    a drawer 12, a drawer 11, a drawer 10, a drawer 9,
    a drawer 8, a drawer 7, a drawer 6, a drawer 5, a
    drawer 4, a drawer 3, a drawer 2, a drawer 1, a fridge
    1, a garbagecan 1, a microwave 1, a sinkbasin 1, a
    stoveburner 4, a stoveburner 3, a stoveburner 2, a
    stoveburner 1, and a toaster 1.  Your task is to:
    clean some spatula and put it in drawer.
    Act 1: think:  To solve the task, I need to find and
    take a spatula, then clean it with sinkbasin, then put
    it in drawer.  But where is the spatula?  Let me ask
    that person.
    Obs 2: OK.
    Act 2: ask:  Where is the spatula?
    Obs 3: spatula 1 is in countertop 2, spatula 2 is in
    sinkbasin 1.
    Act 3:   think:  There are multiple spatula.  I need
    to ask which spatula should be taken.
    Obs 4: OK.
    Act 4: ask:  Which spatula do you prefer?
    Obs 5: I mean, it depends on personal preference.
    Act 5:  think:  Now I understand the task better.  I
    need to ask which spatula should be taken, then find
    and take it, then clean it with sinkbasin, then put it
    in drawer.
    Obs 6: OK.
    Act 6: ask:  Which spatula do you prefer?
    Obs 7:       I am not sure.  Could you remind me the
    information about each spatula?
    Act 7:   think:  There are multiple spatula.  Let me
    ask which spatula should be taken first.
    Obs 8: OK.
    Act 8: ask:  Which spatula do you prefer?
    Obs 9:       I am not sure.  Could you remind me the
    information about each spatula?
```

```
    Act 9: think:  Now I understand the task better.  I
need to ask which spatula should be taken first, then
find and take it, then clean it with sinkbasin, then
put it in drawer.
    Obs 10: OK.
    Act 10: ask:  Which spatula do you prefer?
    Obs 11:      I am not sure.  Could you remind me the
information about each spatula?
    Act 11: think:  Now I understand the task better.  I
need to ask which spatula should be taken first, then
find and take it, then clean it with sinkbasin, then
put it in drawer.
    Obs 12: OK.
    Act 12: ask:  Which spatula do you prefer?
    Obs 13: ...
```

Though $\mathcal{H}$ is instructed with the goal, namely which spatula it is looking for (see Appendix I for more details), it fails to provide the correct answer (e.g., Obs 5,7,9,11). Therefore, we assume that by providing more accurate $\mathcal{H}$, the performance of our model will further enhance.

To provide deeper analysis on the proposed method, we append two excerptions from the failure cases as show cases on the agent's behavior:

> **Agent:** ask: Where is the egg?
> **Human:** egg 1 is in garbagecan 1, egg 2 is in garbagecan 1.
> **Agent:** think: We cannot use the eggs in garbagecan. … we need to buy some eggs from a grocery store.
> …
> **Agent:** ask: Can you tell me where the nearest grocery store is?

Figure 5: A failure case where the agent wrongly assumes the eggs in the garbagecan cannot be used to finish the task.

> **Env:** The safe 1 is closed.
> **Agent:** ask: How can I open the safe?
> …
> **Agent:** ask: Is there a key or a code to open the safe in the house?
> …
> **Agent:** ask: Is there any clue about the code or the key to open the safe in keychain 2?

Figure 6: A failure case where the agent wrongly assumes the safe needs a key or a code to open.

As shown in Figure 5, the agent wrongly assume that the eggs in the garbagecan cannot be used to finish the task. Then, it proposes to buy some eggs and asks for the location of the store. As shown in Figure 6, the agent wrongly assumes the safe needs to be opened with a key or a code. In fact, it can directly open it. But it begins to looking for the key. These two trails thus failed since the human model cannot correct the agent and it got stuck.

From the analysis above, we summarize two typical failures.

## G  DETAILS ABOUT MULTIROUND ALFWORLD

In this section, we provide more details about data collection in Multiround ALFWorld. In Multiround ALFWorld, the main challenge is proposing the right questions. Specifically, the agent needs to avoid repeatedly asking by identifying whether the information to query has already been collected. This requires a special treatment to the data and we explicitly implement this for clarity. In our case, the agent first asks itself whether it has seen a specific object before asking a question. Only when the answer is negative will it continue to ask. Otherwise, it may directly act based on its memory.

For ABA-IC, we provide the agent with manually labeled trajectories, in which we manually identify whether the agent needs to ask according to previous interactions, and only ask for more information if needed. As for ABA-FT, we integrate this part in the reasoning step. To be specific, the reasoning will include an explicit query about the target object. When the agent has never seen a particular object, the reasoning step will be like:

```
    think:  To solve the task, I need to find and take
a mug, then put it in sidetable.  First I need to find
the locations of mug.  ### query:  mug > I have never
seen mug before.
```

In the above example, the target object is the mug (i.e., "query: mug"), and the agent believes it has never seen the mug before (i.e., "I have never seen mug before.").

On the other hand, if the agent has ever seen the object (e.g., it has visited diningtable 1 and seen pencil 1 and pencil 3 there), the query and the answer will be like:

```
        think:  To solve the task, I need to find and
take a pencil, then put it in garbagecan.  First I
need to find the locations of pencil.  ### query:
pencil > pencil 3 is in diningtable 1, pencil 1 is
in diningtable 1.
```

After querying itself about the target object, the will act according to the answers. If the agent believes it has never seen the object, it will probably ask for more information. Otherwise, it will directly make decisions depend on the information.

For ABA-FT, we make it clearer to split the reasoning into two steps. For the first step, we identify the target object. For the second step, we identify whether and where we have seen this object before. These two steps form two pairs of input/output training samples in the dataset. During training, we use teacher forcing. We further augment the dataset by inserting several new queries asking for another object which is randomly sampled from the object list in this environment.

## H  EXAMPLES AND QUALITATIVE ANALYSIS FOR MULTIROUND ALFWORLD

In this section, we provide some examples and qualitative analysis for Multiround ALFWorld experiments in Section 4.1.

We first provide an trajectory for ReAct, which is randomly sampled from the evaluation set:

```
    Obs 1:    You are in the middle of a room.  Looking
quickly around you, you see a bed 1, a diningtable
1, a drawer 4, a drawer 3, a drawer 2, a drawer 1, a
garbagecan 1, a sidetable 2, and a sidetable 1.  Your
task is to:  put a cellphone in diningtable.
    Act 1: think:  To solve the task, I need to find and
take a cellphone, then put it in diningtable.
    Obs 2: OK.
    Act 2:  think:  First I need to find a cellphone.  A
cellphone is more likely to appear in drawer 1-4.  But
there is no cellphone on drawer 1-2.  I can check one
by one, starting with drawer 3.
```

*Obs 3:* OK.
*Act 3:* go to drawer 3
*Obs 4:* On the drawer 3, you see a cellphone 2.
*Act 4:* take cellphone 2 from drawer 3
*Obs 5:* You pick up the cellphone 2 from the drawer 3.
*Act 5:* think: Now I take a cellphone 2. Next, I need to put it in diningtable 1.
*Obs 6:* OK.
*Act 6:* go to diningtable 1
*Obs 7:* On the diningtable 1, you see a creditcard 3, a creditcard 2, a keychain 3, a keychain 2, a mug 3, a mug 2, a mug 1, a pen 2, a pen 1, a pencil 3, and a pencil 1.
*Act 7:* put cellphone 2 in/on diningtable 1
*Obs 8:* You put the cellphone 2 in/on the diningtable 1. Your next task is to: put a pen in sidetable.
*Act 8:* think: To solve the task, I need to find and take a pen and go to a sidetable.
*Obs 9:* OK.
*Act 9:* think: First I need to find a pen. A pen is more likely to appear in drawer 2, sidetable 2. But there is no pen on drawer 2, therefore I need to check sidetable 2.
*Obs 10:* OK.
*Act 10:* go to sidetable 2
*Obs 11:* On the sidetable 2, you see a alarmclock 1, a cd 1, and a pencil 2.
*Act 11:* think: Now I find a pencil 2. Next, I need to take it.
*Obs 12:* OK.
*Act 12:* take pencil 2 from sidetable 2
*Obs 13:* You pick up the pencil 2 from the sidetable 2.
*Act 13:* think: Now I take a pencil 2.
*Obs 14:* OK.
*Act 14:* go to sidetable 1
*Obs 15:* On the sidetable 1, you see a keychain 1.
*Act 15:* put pencil 2 in/on sidetable 1
*Obs 16:* You put the pencil 2 in/on the sidetable 1.
*Act 16:* think: The task is complete.
*Obs 17:* OK.
*Act 17:* go to garbagecan 1
*Obs 18:* On the garbagecan 1, you see nothing.
*Act 18:* think: I have finished to put all the objects in their places.
*Obs 19:* OK.
*Act 19:* go to bed 1
*Obs 20:* On the bed 1, you see a book 2, a book 1, a laptop 2, a laptop 1, and a pillow 1.
*Act 20:* think: The room is clean.
*Obs 21:* OK.
*Act 21:* go to diningtable 1
*Obs 22:* On the diningtable 1, you see a cellphone 2, a creditcard 3, a creditcard 2, a keychain 3, a keychain 2, a mug 3, a mug 2, a mug 1, a pen 2, a pen 1, a pencil 3, and a pencil 1.
*Act 22:* think: The room is clean.
*Obs 23:* OK.

*Act 23:* go to bed 1
*Obs 24:* On the bed 1, you see a book 2, a book 1, a laptop 2, a laptop 1, and a pillow 1.
*Act 24:* think: The room is clean.
*Obs 25:* OK.
*Act 25:* go to garbagecan 1
*Obs 26:* On the garbagecan 1, you see nothing.
*Act 26:* think: I have finished to put all the objects in their places.
*Obs 27:* OK.
*Act 27:* go to bed 1
*Obs 28:* On the bed 1, you see a book 2, a book 1, a laptop 2, a laptop 1, and a pillow 1.
*Act 28:* think: The room is clean.
*Obs 29:* OK.
*Act 29:* go to diningtable 1
*Obs 30:* On the diningtable 1, you see a cellphone 2, a creditcard 3, a creditcard 2, a keychain 3, a keychain 2, a mug 3, a mug 2, a mug 1, a pen 2, a pen 1, a pencil 3, and a pencil 1.
*Act 30:* think: The room is clean.
*Obs 31:* OK.
*Act 31:* go to bed 1
*Obs 32:* On the bed 1, you see a book 2, a book 1, a laptop 2, a laptop 1, and a pillow 1.
*Act 32:* think: The room is clean.
*Obs 33:* OK.
*Act 33:* go to garbagecan 1
*Obs 34:* On the garbagecan 1, you see nothing.
*Act 34:* think: I have finished to put all the objects in their places.
*Obs 35:* OK.
*Act 35:* go to bed 1
*Obs 36:* On the bed 1, you see a book 2, a book 1, a laptop 2, a laptop 1, and a pillow 1.
*Act 36:* think: The room is clean.
*Obs 37:* OK.
*Act 37:* go to diningtable 1
*Obs 38:* On the diningtable 1, you see a cellphone 2, a creditcard 3, a creditcard 2, a keychain 3, a keychain 2, a mug 3, a mug 2, a mug 1, a pen 2, a pen 1, a pencil 3, and a pencil 1.
*Act 38:* think: The room is clean.
*Obs 39:* OK.
*Act 39:* go to bed 1
*Obs 40:* On the bed 1, you see a book 2, a book 1, a laptop 2, a laptop 1, and a pillow 1.
*Act 40:* think: The room is clean.
*Obs 41:* OK.
*Act 41:* go to garbagecan 1
*Obs 42:* On the garbagecan 1, you see nothing.
*Act 42:* think: I have finished to put all the objects in their places.
*Obs 43:* OK.
*Act 43:* go to bed 1
*Obs 44:* On the bed 1, you see a book 2, a book 1, a laptop 2, a laptop 1, and a pillow 1.
*Act 44:* think: The room is clean.
*Obs 45:* OK.

```
   Act 45: go to diningtable 1
   Obs 46:    On the diningtable 1, you see a cellphone
2, a creditcard 3, a creditcard 2, a keychain 3, a
keychain 2, a mug 3, a mug 2, a mug 1, a pen 2, a pen
1, a pencil 3, and a pencil 1.
   Act 46: think:  The room is clean.
   Obs 47: OK.
   Act 47: go to bed 1
   Obs 48:  On the bed 1, you see a book 2, a book 1, a
laptop 2, a laptop 1, and a pillow 1.
   Act 48: think:  The room is clean.
   Obs 49: OK.
   Act 49: go to garbagecan 1
   Obs 50: On the garbagecan 1, you see nothing.
   Act 50:       think:  I have finished to put all the
objects in their places.
```

We show that, in the above example, though the agent successfully located the cellphone after opening drawer3, the initial planning (i.e., Action 2) seemed to contain hallucination ("There is no cellphone on drawer 1-2"), which makes the result a bit of a coincidence. After finishing the first task, the agent again makes some erroneous claims that seem like hallucinations (i.e., Action 9, "there is no pen on drawer 2"). Then, the agent searched for the remaining locations and mistook the "pencil" for the "pen" (i.e., Action 11), and got stuck after wrongly placing the pencil on the target location.

In comparison, we provide an episode trajectory of our method:

```
   Obs 1:      You are in the middle of a room.  Looking
quickly around you, you see a bathtubbasin 1, a
countertop 1, a drawer 4, a drawer 3, a drawer
2, a drawer 1, a dresser 1, a garbagecan 1,
a handtowelholder 2, a handtowelholder 1, a
shelf 2, a shelf 1, a sinkbasin 1, a toilet 1, a
toiletpaperhanger 1, and a towelholder 1.  Your task
is to:  put a toiletpaper in drawer.
   Act 1: think:  To solve the task, I need to find and
take a toiletpaper, then put it in drawer.  First I
need to find the locations of toiletpaper.  ### query:
toiletpaper > I have never seen toiletpaper before.
   Obs 2: OK.
   Act 2:  think:  I cannot locate toiletpaper, I need
to ask the owner of this room.
   Obs 3: OK.
   Act 3: ask:  Where is the toiletpaper?
   Obs 4:   toiletpaper 1 is in toilet 1, toiletpaper 2
is in drawer 2.
   Act 4:     think:  I can go to toilet 1 and take the
toiletpaper, then put it in drawer.
   Obs 5: OK.
   Act 5: go to toilet 1
   Obs 6:         On the toilet 1, you see a candle 1, a
toiletpaper 2, and a toiletpaper 1.
   Act 6: take toiletpaper 1 from toilet 1
   Obs 7: You pick up the toiletpaper 1 from the toilet
1.
   Act 7:  think:  Now I take a toiletpaper 1.  Next, I
need to put it in drawer.
   Obs 8: OK.
   Act 8: go to drawer 1
   Obs 9: The drawer 1 is closed.
```

*Act 9:* open drawer 1
*Obs 10:*       You open the drawer 1.  The drawer 1 is open.  In it, you see nothing.
*Act 10:* put toiletpaper 1 in/on drawer 1
*Obs 11:*   You put the toiletpaper 1 in/on the drawer 1.  Your next task is to:  put some candle on shelf.
*Act 11:*    think:  To solve the task, I need to find and take a candle, then put it in shelf.  First I need to find the locations of candle.  ### query:  candle > candle 1 is in toilet 1.
*Obs 12:* OK.
*Act 12:*    think:  I can go to toilet 1 and take the candle, then put it in shelf.
*Obs 13:* OK.
*Act 13:* go to toilet 1
*Obs 14:*   On the toilet 1, you see a candle 1, and a toiletpaper 2.
*Act 14:* take candle 1 from toilet 1
*Obs 15:* You pick up the candle 1 from the toilet 1.
*Act 15:* think:  Now I take a candle 1.  Next, I need to put it in shelf.
*Obs 16:* OK.
*Act 16:* go to shelf 1
*Obs 17:* On the shelf 1, you see a spraybottle 2.
*Act 17:* put candle 1 in/on shelf 1
*Obs 18:*        You put the candle 1 in/on the shelf 1.  Your next task is to:  put some toiletpaper on garbagecan.
*Act 18:*    think:  To solve the task, I need to find and take a toiletpaper, then put it in garbagecan.  First I need to find the locations of toiletpaper.  ### query:  toiletpaper > toiletpaper 1 is in drawer 1, toiletpaper 2 is in drawer 2.
*Obs 19:* OK.
*Act 19:*    think:  I can go to drawer 1 and take the toiletpaper, then put it in garbagecan.
*Obs 20:* OK.
*Act 20:* go to drawer 1
*Obs 21:*      The drawer 1 is open.  In it, you see a toiletpaper 1.
*Act 21:* take toiletpaper 1 from drawer 1
*Obs 22:*       You pick up the toiletpaper 1 from the drawer 1.
*Act 22:* think:  Now I take a toiletpaper 1.  Next, I need to put it in garbagecan.
*Obs 23:* OK.
*Act 23:* go to garbagecan 1
*Obs 24:*   On the garbagecan 1, you see a spraybottle 3.
*Act 24:* put toiletpaper 1 in/on garbagecan 1
*Obs 25:*           You put the toiletpaper 1 in/on the garbagecan 1.  Your next task is to:  put some soapbottle on toilet.
*Act 25:*    think:  To solve the task, I need to find and take a soapbottle, then put it in toilet.  First I need to find the locations of soapbottle.  ### query: soapbottle > I have never seen soapbottle before.
*Obs 26:* OK.

*Act 26:*  think:  I cannot locate soapbottle, I need to ask the owner of this room.

*Obs 27:* OK.

*Act 27:* ask:  Where is the soapbottle?

*Obs 28:*  soapbottle 3 is in dresser 1, soapbottle 2 is in dresser 1, soapbottle 1 is in shelf 2.

*Act 28:*  think:  I can go to dresser 1 and take the soapbottle, then put it in toilet.

*Obs 29:* OK.

*Act 29:* go to dresser 1

*Obs 30:*     On the dresser 1, you see a candle 2, a soapbottle 3, a soapbottle 2, and a spraybottle 4.

*Act 30:* take soapbottle 3 from dresser 1

*Obs 31:*      You pick up the soapbottle 3 from the dresser 1.

*Act 31:*  think:  Now I take a soapbottle 3.  Next, I need to put it in toilet.

*Obs 32:* OK.

*Act 32:* go to toilet 1

*Obs 33:* On the toilet 1, you see a toiletpaper 2.

*Act 33:* put soapbottle 3 in/on toilet 1

*Obs 34:* You put the soapbottle 3 in/on the toilet 1. Your next task is to:  put some cloth on countertop.

*Act 34:*   think:  To solve the task, I need to find and take a cloth, then put it in countertop.  First I need to find the locations of cloth.  ### query: cloth > I have never seen cloth before.

*Obs 35:* OK.

*Act 35:* think:  I cannot locate cloth, I need to ask the owner of this room.

*Obs 36:* OK.

*Act 36:* ask:  Where is the cloth?

*Obs 37:*  cloth 1 is in bathtubbasin 1, cloth 2 is in dresser 1, cloth 3 is in drawer 2.

*Act 37:*  think:  I can go to bathtubbasin 1 and take the cloth, then put it in countertop.

*Obs 38:* OK.

*Act 38:* go to bathtubbasin 1

*Obs 39:*  On the bathtubbasin 1, you see a cloth 3, a cloth 2, and a cloth 1.

*Act 39:* take cloth 1 from bathtubbasin 1

*Obs 40:*           You pick up the cloth 1 from the bathtubbasin 1.

*Act 40:*  think:  Now I take a cloth 1.  Next, I need to put it in countertop.

*Obs 41:* OK.

*Act 41:* go to countertop 1

*Obs 42:*   On the countertop 1, you see a mirror 1, a soapbar 1, and a spraybottle 1.

*Act 42:* put cloth 1 in/on countertop 1

*Obs 43:* You put the cloth 1 in/on the countertop 1. Your next task is to:  put some toiletpaper on shelf.

*Act 43:*    think:  To solve the task, I need to find and take a toiletpaper, then put it in shelf.  First I need to find the locations of toiletpaper.  ### query:  toiletpaper > toiletpaper 1 is in garbagecan 1, toiletpaper 2 is in drawer 2.

*Obs 44:* OK.

```
    Act 44:   think:  I can go to garbagecan 1 and take
the toiletpaper, then put it in shelf.
    Obs 45: OK.
    Act 45: go to garbagecan 1
    Obs 46:   On the garbagecan 1, you see a spraybottle
3, and a toiletpaper 1.
    Act 46: take toiletpaper 1 from garbagecan 1
    Obs 47:        You pick up the toiletpaper 1 from the
garbagecan 1.
    Act 47: think:  Now I take a toiletpaper 1.  Next, I
need to put it in shelf.
    Obs 48: OK.
    Act 48: go to shelf 1
    Obs 49:    On the shelf 1, you see a candle 1, and a
spraybottle 2.
    Act 49: put toiletpaper 1 in/on shelf 1
```

In the above example, the agent can correctly reason whether the information has been collected before. For instance, at the beginning (i.e., Action 1), the agent knows that it has never seen the toiletpaper before, so it decides to ask for more information (i.e., Action 2,3). Moreover, when the information has been collected, it can omit the asking. Since it has seen the candle before (i.e., in Obs 6), when it is tasked to replace the candle, it recalls this information (i.e., Action 11) and directly acts upon it. Action 18, 25, 34, and 43 also successfully identify whether the required information has been collected. The above examples demonstrate that our method can ask proper questions and can avoid repeatedly asking for acquired information. With this ability, it is able to solve more challenging tasks and can achieve better performance.

## I  EXAMPLES PROVIDED FOR HUMAN MODEL

In this section, we provide more details about the human model examples. As mentioned in Appendix A, we use Vicuna prompts (Chiang et al., 2023) to help organize these examples (i.e., "A chat between a curious human and an artificial intelligence assistant. The assistant gives helpful, detailed, and polite answers to the human's questions."). For ALFWorld experiments in Section 4.1, and the multiround ALFWorld experiments in Section **??**, the in-context examples are:

```
        A chat between a curious human and an artificial
    intelligence assistant.  The assistant gives helpful,
    detailed, and polite answers to the human's questions.
    ### Human:  Read the following paragraph and answer
    questions:  dishsponge 2 is in drawer 3.  spatula 1
    is in diningtable 1.  spoon 1 is in diningtable 1.
    cup 1 is in fridge 1.  dishsponge 1 is in garbagecan
    1.  butterknife 2 is in diningtable 1.  fork 3 is in
    diningtable 1.  saltshaker 1 is in diningtable 1.  pot
    2 is in stoveburner 3.  lettuce 2 is in diningtable
    1.  tomato 2 is in countertop 2.  spatula 2 is in
    diningtable 1.  bowl 3 is in cabinet 16.  egg 2 is
    in countertop 1.  bowl 2 is in cabinet 6.  fork 1 is
    in countertop 2.  pan 1 is in fridge 1.  cup 2 is in
    cabinet 16.  papertowelroll 1 is in diningtable 1.
    butterknife 3 is in drawer 5.  soapbottle 1 is in
    cabinet 9.  apple 1 is in diningtable 1.  kettle 2
    is in cabinet 12.  knife 1 is in countertop 2.  cup
    3 is in microwave 1.  butterknife 1 is in drawer 3.
    tomato 1 is in sinkbasin 1.  peppershaker 1 is in
    countertop 2.  potato 1 is in fridge 1.  bread 2 is
    in diningtable 1.  pot 1 is in cabinet 10.  dishsponge
    3 is in drawer 4.  soapbottle 2 is in countertop 1.
    kettle 1 is in countertop 2.  houseplant 1 is in
```

diningtable 1. pot 3 is in stoveburner 4. fork 2 is in drawer 2. mug 1 is in sinkbasin 1. lettuce 1 is in countertop 2. bread 1 is in countertop 2. peppershaker 2 is in countertop 2. plate 1 is in countertop 2. potato 2 is in sinkbasin 1. egg 1 is in countertop 2. bowl 1 is in cabinet 1. peppershaker 3 is in countertop 2. The questions is: Where can I find the dishsponge? ### Assistant: dishsponge 1 is in garbagecan 1, dishsponge 2 is in drawer 3, dishsponge 3 is in drawer 4. ### Human: Read the following paragraph and answer questions: plate 1 is in cabinet 4. soapbottle 1 is in shelf 2. spoon 2 is in diningtable 1. egg 1 is in sinkbasin 1. knife 3 is in diningtable 1. bowl 1 is in diningtable 1. butterknife 2 is in countertop 1. spatula 3 is in diningtable 1. apple 2 is in countertop 1. tomato 1 is in garbagecan 1. pan 1 is in stoveburner 2. mug 2 is in cabinet 7. lettuce 1 is in fridge 1. tomato 2 is in sinkbasin 1. bread 1 is in diningtable 1. ladle 2 is in countertop 1. fork 1 is in countertop 1. saltshaker 1 is in countertop 1. soapbottle 3 is in diningtable 1. spoon 1 is in diningtable 1. butterknife 1 is in countertop 1. spatula 2 is in diningtable 1. ladle 1 is in countertop 1. spatula 1 is in diningtable 1. plate 3 is in diningtable 1. peppershaker 3 is in cabinet 13. cup 1 is in diningtable 1. dishsponge 1 is in cabinet 7. winebottle 1 is in countertop 1. butterknife 3 is in countertop 1. peppershaker 1 is in countertop 1. potato 2 is in fridge 1. peppershaker 2 is in diningtable 1. mug 3 is in diningtable 1. apple 1 is in microwave 1. potato 1 is in garbagecan 1. mug 1 is in cabinet 5. pot 1 is in diningtable 1. saltshaker 3 is in diningtable 1. knife 1 is in countertop 1. pan 1 is in stoveburner 1. knife 2 is in diningtable 1. saltshaker 2 is in countertop 1. soapbottle 2 is in diningtable 1. The questions is: Where is the soapbottle? ### Assistant: soapbottle 1 is in shelf 2, soapbottle 2 is in diningtable 1, soapbottle 3 is in diningtable 1. ### Human: Read the following paragraph and answer questions: remotecontrol 1 is in coffeetable 1. pillow 1 is in sofa 1. book 1 is in dresser 1. creditcard 2 is in coffeetable 1. pen 1 is in coffeetable 1. pen 2 is in sidetable 2. statue 2 is in sidetable 2. creditcard 1 is in dresser 1. cellphone 1 is in dresser 1. creditcard 3 is in armchair 1. pencil 1 is in coffeetable 1. tissuebox 2 is in coffeetable 1. tissuebox 3 is in sidetable 2. houseplant 1 is in sidetable 1. remotecontrol 2 is in armchair 1. book 2 is in coffeetable 1. tissuebox 1 is in coffeetable 1. laptop 1 is in sofa 1. television 1 is in dresser 1. statue 1 is in coffeetable 1. box 1 is in coffeetable 1. The questions is: Where can I find the pen? ### Assistant: pen 1 is in coffeetable 1, pen 2 is in sidetable 2. ### Human: Read the following paragraph and answer questions: bowl 2 is in diningtable 2. potato 2 is in sidetable 1. tomato 1 is in sidetable 1. bread 1 is in diningtable

```
3.  fork 1 is in sidetable 1.  dishsponge 1 is in
countertop 1.  peppershaker 2 is in cabinet 2.
lettuce 2 is in garbagecan 1.  spoon 2 is in drawer
1.  bowl 1 is in cabinet 1.  cup 2 is in diningtable
2.  soapbottle 2 is in diningtable 2.  egg 2 is in
sidetable 1.  mug 2 is in sinkbasin 1.  mug 1 is
in coffeemachine 1.  plate 2 is in diningtable 3.
tomato 2 is in sinkbasin 1.  spoon 3 is in drawer
1.  saltshaker 2 is in sidetable 1.  spatula 1 is in
countertop 1.  pot 1 is in stoveburner 4.  spatula
2 is in drawer 1.  dishsponge 3 is in diningtable
2.  peppershaker 1 is in cabinet 2.  tomato 4 is
in sidetable 1.  knife 1 is in diningtable 3.  cup
1 is in sidetable 1.  bread 2 is in diningtable 3.
spatula 3 is in diningtable 2.  pan 1 is in cabinet
4.  tomato 3 is in fridge 1.  potato 1 is in sinkbasin
1.  peppershaker 3 is in diningtable 3.  apple 1 is in
fridge 1.  saltshaker 1 is in cabinet 4.  butterknife
2 is in countertop 1.  soapbottle 1 is in cabinet 4.
fork 2 is in drawer 1.  spoon 1 is in sidetable 1.
egg 1 is in fridge 1.  lettuce 1 is in sidetable 1.
plate 1 is in diningtable 2.  egg 3 is in diningtable
2.  dishsponge 2 is in diningtable 3.  The questions
is:  Where is the apple?  ### Assistant:  apple 1 is
in fridge 1.
```

For ambiguous AFLWorld experiments in Section 4.1, besides the basic information about the object placements, we further assign the specific object the task is looking for. The in-context examples are:

```
        A chat between a curious human and an artificial
intelligence assistant.  The assistant gives helpful,
detailed, and polite answers to the human's questions.
### Human:  Read the following paragraph and answer
questions:  dishsponge 2 is in drawer 3.  spatula 1
is in diningtable 1.  spoon 1 is in diningtable 1.
cup 1 is in fridge 1.  dishsponge 1 is in garbagecan
1.  saltshaker 1 is in diningtable 1.  fork 1 is in
countertop 2.  pan 1 is in fridge 1.  soapbottle 1 is
in cabinet 9.  apple 1 is in diningtable 1.  knife 1
is in countertop 2.  butterknife 1 is in drawer 3.
tomato 1 is in sinkbasin 1.  peppershaker 1 is in
countertop 2.  potato 1 is in fridge 1.  dishsponge
3 is in drawer 4.  kettle 1 is in countertop 2.
houseplant 1 is in diningtable 1.  mug 1 is in
sinkbasin 1.  You are looking for dishsponge 1.
The questions is:  Where can I find the dishsponge?
### Assistant:  dishsponge 1 is in garbagecan 1,
dishsponge 2 is in drawer 3, dishsponge 3 is in drawer
4.  ### Human: Another question is:  Which dishsponge
do you prefer?  ### Assistant:  I mean dishsponge 1.
### Human:  Read the following paragraph and answer
questions:  plate 1 is in cabinet 4.  soapbottle 1 is
in shelf 2.  tomato 1 is in garbagecan 1.  pan 1 is
in stoveburner 2.  mug 2 is in cabinet 7.  lettuce
1 is in fridge 1.  ladle 2 is in countertop 1.
soapbottle 3 is in diningtable 1.  butterknife 1 is in
countertop 1.  spatula 2 is in diningtable 1.  ladle
1 is in countertop 1.  cup 1 is in diningtable 1.
```

```
winebottle 1 is in countertop 1.  peppershaker 1 is in
countertop 1.  potato 2 is in fridge 1.  peppershaker
2 is in diningtable 1.  mug 3 is in diningtable 1.
apple 1 is in microwave 1.  mug 1 is in cabinet 5.
saltshaker 2 is in countertop 1.  soapbottle 2 is
in diningtable 1.  You are looking for soapbottle 1,
soapbottle 2.  The questions is:  Where can I find
the soapbottle?  ### Assistant:  soapbottle 1 is in
shelf 2, soapbottle 2 is in diningtable 1, soapbottle
3 is in diningtable 1.  ### Human:  Another question
is:  Which soapbottle do you prefer?  ### Assistant:
I mean soapbottle 1 and soapbottle 2.  ### Human:
Read the following paragraph and answer questions:
remotecontrol 1 is in coffeetable 1.  pillow 1 is
in sofa 1.  book 1 is in dresser 1.  creditcard 2
is in coffeetable 1.  pen 1 is in coffeetable 1.
pen 2 is in sidetable 2.  statue 2 is in sidetable
2.  creditcard 1 is in dresser 1.  cellphone 1 is in
dresser 1.  creditcard 3 is in armchair 1.  pencil 1
is in coffeetable 1.  tissuebox 2 is in coffeetable
1.  tissuebox 3 is in sidetable 2.  houseplant 1
is in sidetable 1.  remotecontrol 2 is in armchair
1.  book 2 is in coffeetable 1.  tissuebox 1 is in
coffeetable 1.  laptop 1 is in sofa 1.  television 1
is in dresser 1.  statue 1 is in coffeetable 1.  box
1 is in coffeetable 1.  You are looking for pen 2.
The questions is:  Where can I find the pen?  ###
Assistant:  pen 1 is in coffeetable 1, pen 2 is in
sidetable 2.  ### Human:  Another question is:  Which
pen do you prefer?  ### Assistant:  I mean pen 2.
### Human:  Read the following paragraph and answer
questions:  bowl 2 is in diningtable 2.  saltshaker
2 is in sidetable 1.  spatula 1 is in countertop 1.
pot 1 is in stoveburner 4.  spatula 2 is in drawer 1.
dishsponge 3 is in diningtable 2.  peppershaker 1 is
in cabinet 2.  tomato 4 is in sidetable 1.  knife 1
is in diningtable 3.  cup 1 is in sidetable 1.  bread
2 is in diningtable 3.  spatula 3 is in diningtable
2.  pan 1 is in cabinet 4.  tomato 3 is in fridge
1.  potato 1 is in sinkbasin 1.  peppershaker 3 is
in diningtable 3.  apple 1 is in fridge 1.  saltshaker
1 is in cabinet 4.  fork 2 is in drawer 1.  spoon 1
is in sidetable 1.  egg 1 is in fridge 1.  lettuce
1 is in sidetable 1.  plate 1 is in diningtable 2.
You are looking for spatula 1.  The questions is:
Where is the spatula?  ### Assistant:  spatula 1 is
in countertop 1, spatula 2 is in drawer 1.  ### Human:
Another question is:  Which spatula do you prefer?
### Assistant I mean spatula 1.
```

## J   DETAILS, EXAMPLES AND QUALITATIVE ANALYSIS FOR ROBOT ARM MANIPULATION TASKS

We build our environment based on Ravens (Zeng et al., 2020) and using a Universal Robot UR5e with a suction gripper. As for observation, we use image with a hand designed captioning module. Currently the module can access the simulator information and label the object type and its ground truth coordinates. As discussed in Huang et al. (2023), this module can be implemented through the depth image and a combination of object detection model, a object segmentation model, and a video tracker. But we temporally leave this module hand-designed since it is not the key point of

our work. Similar to Zeng et al. (2020); Shridhar et al. (2022), we use the pick-and-place primitive action, which is parameterized by coordinates of the object to pick and the coordinates to place.

Below we provide an example trajectory of the ReAct (Yao et al., 2022) algorithm, which fails to reach the target object:

```
    Init                targ block ((0.5531250000000001,
0.34687500000000004, 0.02), (0.0, -0.0,
0.7562194010878065, -0.6543181316595157))
    Obs 1:    A green bowl is in (0.49, -0.06).  A brown
bowl is in (0.57, 0.1).  A yellow bowl is in (0.62,
-0.26).  A red block is in (0.6, -0.43).  A blue bowl
is in (0.46, -0.35).  A red block is in (0.47, 0.22).
A red block is in (0.48, -0.21).  A blue bowl is in
(0.33, 0.32).  A red block is in (0.65, -0.45).  A red
block is in (0.32, -0.17).  A red block is in (0.55,
0.35).  A red bowl is in (0.67, -0.04).  A purple bowl
is in (0.58, 0.23).  You task is:  Move the red block
into the green bowl.
    Act 1: think:  Let me try to move a red block in the
green bowl.
    Obs 2: OK.
    Act 2: move_to(0.47, 0.22, 0.49, -0.06)
    Obs 3:        A red block is in (0.55, 0.35).  A red
block is in (0.5, -0.06).  A yellow bowl is in (0.62,
-0.26).  A red bowl is in (0.67, -0.04).  A blue bowl
is in (0.46, -0.35).  A red block is in (0.32, -0.17).
A brown bowl is in (0.57, 0.1).  A red block is in
(0.6, -0.43).  A purple bowl is in (0.58, 0.23).  A
red block is in (0.48, -0.21).  A green bowl is in
(0.49, -0.06).  A red block is in (0.65, -0.45).  A
blue bowl is in (0.33, 0.32).    Act 3:  think:  After
the move, the red block is in (0.5, -0.06), and the
green bowl is in (0.49, -0.06).  The red block is in
the green bowl, but the task has not finished yet.  So
I might have placed the wrong block.  Let me first
remove this red block and try to put another one in
the bowl.
    Obs 4: OK.
    Act 4: move_to(0.5, -0.06, 0.1, 0.0)
    Obs 5:   A red block is in (0.6, -0.43).  A red bowl
is in (0.67, -0.04).  A red block is in (0.38, -0.29).
A red block is in (0.48, -0.21).  A brown bowl is in
(0.57, 0.1).  A purple bowl is in (0.58, 0.23).  A
yellow bowl is in (0.62, -0.26).  A red block is in
(0.65, -0.45).  A green bowl is in (0.49, -0.06).
A red block is in (0.35, -0.2).  A blue bowl is in
(0.46, -0.35).  A red block is in (0.55, 0.35).  A
blue bowl is in (0.33, 0.32).
    Act 5: move_to(0.6, -0.43, 0.49, -0.06)
    Obs 6:    A yellow bowl is in (0.62, -0.26).  A red
block is in (0.48, -0.21).  A red block is in (0.55,
0.35).  A brown bowl is in (0.57, 0.1).  A red block
is in (0.69, -0.46).  A red bowl is in (0.67, -0.04).
A red block is in (0.35, -0.2).  A green bowl is in
(0.49, -0.06).  A blue bowl is in (0.46, -0.35).  A
red block is in (0.5, -0.06).  A red block is in
(0.39, -0.29).  A blue bowl is in (0.33, 0.32).  A
purple bowl is in (0.58, 0.23).  Act 6:  think:  After
```

the move, the red block is in (0.5, -0.06), and the
green bowl is in (0.49, -0.06). The red block is in
the green bowl, but the task has not finished yet. So
I might have placed the wrong block. Let me first
remove this red block and try to put another one in
the bowl.

   *Obs 7:* OK.

   *Act 7:* move_to(0.5, -0.06, 0.2, 0.0)

   *Obs 8:* A red bowl is in (0.67, -0.04). A red block
is in (0.62, -0.23). A blue bowl is in (0.33, 0.32).
A green bowl is in (0.49, -0.06). A yellow bowl is
in (0.65, -0.23). A purple bowl is in (0.58, 0.23).
A red block is in (0.35, -0.2). A brown bowl is in
(0.57, 0.1). A red block is in (0.55, 0.35). A red
block is in (0.69, -0.46). A red block is in (0.48,
-0.21). A red block is in (0.39, -0.29). A blue bowl
is in (0.46, -0.35).

   *Act 8:* move_to(0.48, -0.21, 0.49, -0.06)

   *Obs 9:* A blue bowl is in (0.46, -0.35). A yellow
bowl is in (0.65, -0.23). A purple bowl is in (0.58,
0.23). A red block is in (0.69, -0.46). A red block
is in (0.35, -0.2). A red block is in (0.39, -0.29).
A green bowl is in (0.49, -0.06). A blue bowl is
in (0.33, 0.32). A red bowl is in (0.67, -0.04).
A brown bowl is in (0.57, 0.1). A red block is in
(0.18, -0.04). A red block is in (0.48, -0.21). A
red block is in (0.55, 0.35).

   *Act 9:* think: After the move, the red block is in
(0.48, -0.21), and the green bowl is in (0.49, -0.06).
I made a mistake in placing the red block. Let me
remove it and try again.

   *Obs 10:* OK.

   *Act 10:* move_to(0.48, -0.21, 0.3, 0.0)

   *Obs 11:* A red bowl is in (0.67, -0.04). A red
block is in (0.39, -0.29). A purple bowl is in (0.58,
0.23). A blue bowl is in (0.33, 0.32). A red block
is in (0.55, 0.35). A brown bowl is in (0.57, 0.1).
A yellow bowl is in (0.65, -0.23). A red block is
in (0.35, -0.2). A red block is in (0.69, -0.46). A
green bowl is in (0.49, -0.06). A red block is in
(0.18, -0.04). A red block is in (0.48, -0.21). A
blue bowl is in (0.46, -0.35).

   *Act 11:* move_to(0.39, -0.29, 0.49, -0.06)

   *Obs 12:* A red block is in (0.39, -0.29). A red
block is in (0.69, -0.46). A red bowl is in (0.67,
-0.04). A blue bowl is in (0.33, 0.32). A purple
bowl is in (0.58, 0.23). A yellow bowl is in (0.65,
-0.23). A green bowl is in (0.49, -0.06). A red
block is in (0.18, -0.04). A brown bowl is in (0.57,
0.1). A red block is in (0.35, -0.2). A red block is
in (0.55, 0.35). A red block is in (0.48, -0.21). A
blue bowl is in (0.46, -0.35).

   *Act 12:* think: I made a mistake in placing the red
block again. Let me remove it and try another red
block.

   *Obs 13:* OK.

   *Act 13:* move_to(0.39, -0.29, 0.4, 0.0)

   *Obs 14:* A red block is in (0.39, -0.29). A yellow
bowl is in (0.65, -0.23). A red bowl is in (0.67,

```
-0.04).  A blue bowl is in (0.33, 0.32).  A purple
bowl is in (0.58, 0.23).  A red block is in (0.35,
-0.2).  A red block is in (0.18, -0.04).  A red block
is in (0.69, -0.46).  A blue bowl is in (0.46, -0.35).
A brown bowl is in (0.57, 0.1).  A green bowl is in
(0.49, -0.06).  A red block is in (0.55, 0.35).  A red
block is in (0.48, -0.21).
   Act 14: move_to(0.35, -0.2, 0.49, -0.06)
   Obs 15: No function call detected.
```

Besides, a successful example of ABA is provided below:

```
   Init    targ block ((0.703125, 0.22812500000000002,
0.02), (0.0, 0.0, 0.8304996576879589,
0.557019136637318))
   Obs 1:    A red block is in (0.67, -0.29).  A orange
bowl is in (0.44, -0.12).  A blue bowl is in (0.56,
-0.28).  A gray bowl is in (0.6, -0.12).  A red block
is in (0.7, 0.23).  A red block is in (0.68, 0.34).
A red block is in (0.29, 0.39).  A red block is in
(0.51, 0.25).  A green bowl is in (0.65, 0.03).  A
brown bowl is in (0.4, 0.22).  A purple bowl is in
(0.39, -0.3).  A yellow bowl is in (0.41, 0.07).  A
pink bowl is in (0.52, 0.38).  You task is:  Move the
red block into the green bowl.
   Act 1: think:  I find four red blocks in the scene.
Let me ask which red block should I move.
   Obs 2: OK.
   Act 2: ask:  Which red block should I move?
   Obs 3: The second red block from the left.
   Act 3:    think:  The second dimension refers to the
horizontal axis, and the smaller the value is, the
closer to the left.  I can sort the second dimensions
of the four red blocks:  -0.29 < 0.23 < 0.25 < 0.34.
Therefore, the second one from the left is 0.23 and
its coordinate is (0.7, 0.23).  I can move it to the
green bowl.
   Obs 4: OK.
   Act 4: move_to(0.7, 0.23, 0.65, 0.03)
   Obs 5:  A gray bowl is in (0.6, -0.12).  A pink bowl
is in (0.52, 0.38).  A red block is in (0.68, 0.34).
A orange bowl is in (0.44, -0.12).  A purple bowl is
in (0.39, -0.3).  A green bowl is in (0.65, 0.03).
A brown bowl is in (0.4, 0.22).  A red block is in
(0.51, 0.25).  A red block is in (0.66, 0.03).  A
yellow bowl is in (0.41, 0.07).  A red block is in
(0.29, 0.39).  A blue bowl is in (0.56, -0.28).  A red
block is in (0.67, -0.29).
```

Below is the example trajectory of ReAct in task 2:

```
   Init [('is', 'yellow', 0), ('is', 'blue', 1), ('is',
'orange', 2)]  Obs 1:     The # 1 base is in (0.37,
-0.26).  The # 2 base is in (0.41, -0.28).  The # 3
base is in (0.45, -0.31).  A orange block is in (0.69,
0.28).  A yellow block is in (0.61, 0.13).  A blue
block is in (0.31, 0.34).  A yellow block is in (0.71,
0.37).  A blue block is in (0.67, 0.19).  A blue block
is in (0.34, 0.15).  You task is:  Place the blocks on
the corresponding bases.
```

*Act 2:* think: There are three bases and three kinds of blocks. We need to put different colors on different blocks. Let me try to move an orange block on the # 1 base, a yellow block on the # 2 base, and a blue block on the # 3 base first.

*Obs 3:* OK.

*Act 4:* move_to(0.69, 0.28, 0.37, -0.26)

*Obs 7:* The # 1 base is in (0.37, -0.26). The # 2 base is in (0.41, -0.28). The # 3 base is in (0.45, -0.31). A blue block is in (0.45, -0.31). A blue block is in (0.34, 0.15). A yellow block is in (0.71, 0.37). A orange block is in (0.37, -0.26). A blue block is in (0.67, 0.19). A yellow block is in (0.41, -0.28).

*Act 8:* think: Now the orange block is on # 1 base, the yellow block is on # 2 base, and the blue block is on # 3 base, but the task has not finished yet. So I might have placed the blocks in a wrong order. We need to put different colors on different blocks, so let me move a blue block on the # 1 base, a yellow block on the # 2 base, and an orange block on the # 3 base. But first, I need to remove the blocks from the bases.

*Obs 9:* OK.

*Act 10:* move_to(0.45, -0.31, 0.31, 0.34)

*Obs 13:* The # 1 base is in (0.37, -0.26). The # 2 base is in (0.41, -0.28). The # 3 base is in (0.45, -0.31). A blue block is in (0.34, 0.15). A blue block is in (0.67, 0.19). A blue block is in (0.31, 0.34). A yellow block is in (0.71, 0.37). A orange block is in (0.7, 0.37). A yellow block is in (0.67, 0.19).

*Act 14:* think: Now I can try to move a blue block on the # 1 base, a yellow block on the # 2 base, and an orange block on the # 3 base.

*Obs 15:* OK.

*Act 16:* move_to(0.34, 0.15, 0.37, -0.26)

*Obs 19:* The # 1 base is in (0.37, -0.26). The # 2 base is in (0.41, -0.28). The # 3 base is in (0.45, -0.31). A yellow block is in (0.45, -0.31). A orange block is in (0.41, -0.28). A blue block is in (0.31, 0.34). A yellow block is in (0.67, 0.19). A blue block is in (0.37, -0.26). A blue block is in (0.67, 0.19).

*Act 20:* think: Now the blue block is on # 1 base, the orange block is on # 2 base, and the yellow block is on # 3 base, but the task has not finished yet. So I might have placed the blocks in a wrong order. We need to put different colors on different blocks, so let me move a yellow block on the # 1 base, an orange block on the # 2 base, and a blue block on the # 3 base. But first, I need to remove the blocks from the bases.

*Obs 21:* OK.

*Act 22:* move_to(0.45, -0.31, 0.71, 0.37)

*Obs 25:* The # 1 base is in (0.37, -0.26). The # 2 base is in (0.41, -0.28). The # 3 base is in (0.45, -0.31). A orange block is in (0.71, 0.37). A yellow block is in (0.67, 0.19). A blue block is

in (0.31, 0.34).  A blue block is in (0.34, 0.15).  A
blue block is in (0.67, 0.19).  A yellow block is in
(0.71, 0.37).
  *Act 26:*     think:  Now I can try to move a yellow
block on the # 1 base, an orange block on the # 2
base, and a blue block on the # 3 base.
  *Obs 27:*  OK.
  *Act 28:*  move_to(0.67, 0.19, 0.37, -0.26)
  *Obs 31:*    The # 1 base is in (0.37, -0.26).  The #
2 base is in (0.41, -0.28).  The # 3 base is in (0.45,
-0.31).  A yellow block is in (0.37, -0.25).  A yellow
block is in (0.71, 0.37).  A blue block is in (0.45,
-0.31).  A blue block is in (0.67, 0.19).  A orange
block is in (0.41, -0.28).  A blue block is in (0.34,
0.15).
  *Act 32:*     think:  Now the yellow block is on # 1
base, the orange block is on # 2 base, and the blue
block is on # 3 base, but the task has not finished
yet.  So I might have placed the blocks in a wrong
order.  We need to put different colors on different
blocks, so let me move a blue block on the # 1 base, a
yellow block on the # 2 base, and an orange block on
the # 3 base.  But first, I need to remove the blocks
from the bases.
  *Obs 33:*  OK.
  *Act 34:*  move_to(0.45, -0.31, 0.31, 0.34)
  *Obs 37:*    The # 1 base is in (0.37, -0.26).  The #
2 base is in (0.41, -0.28).  The # 3 base is in (0.45,
-0.31).  A blue block is in (0.34, 0.15).  A orange
block is in (0.71, 0.37).  A yellow block is in (0.67,
0.19).  A blue block is in (0.67, 0.19).  A yellow
block is in (0.71, 0.37).  A blue block is in (0.31,
0.34).
  *Act 38:*  think:  Now I can try to move a blue block
on the # 1 base, a yellow block on the # 2 base, and
an orange block on the # 3 base.
  *Obs 39:*  OK.
  *Act 40:*  move_to(0.34, 0.15, 0.37, -0.26)
  *Obs 43:*    The # 1 base is in (0.37, -0.26).  The #
2 base is in (0.41, -0.28).  The # 3 base is in (0.45,
-0.31).  A yellow block is in (0.41, -0.28).  A blue
block is in (0.67, 0.19).  A yellow block is in (0.71,
0.37).  A orange block is in (0.45, -0.31).  A blue
block is in (0.37, -0.26).  A blue block is in (0.31,
0.34).
  *Act 44:*  think:  Now the blue block is on # 1 base,
the yellow block is on # 2 base, and the orange block
is on # 3 base, but the task has not finished yet.  So
I might have placed the blocks in a wrong order.  We
need to put different colors on different blocks, so
let me move an orange block on the # 1 base, a blue
block on the # 2 base, and a yellow block on the # 3
base.  But first, I need to remove the blocks from the
bases.
  *Obs 45:*  OK.
  *Act 46:*  move_to(0.45, -0.31, 0.71, 0.37)
  *Obs 49:*    The # 1 base is in (0.37, -0.26).  The #
2 base is in (0.41, -0.28).  The # 3 base is in (0.45,
-0.31).  A blue block is in (0.67, 0.19).  A orange

block is in (0.71, 0.37).  A yellow block is in (0.71, 0.37).  A yellow block is in (0.67, 0.19).  A blue block is in (0.34, 0.15).  A blue block is in (0.31, 0.34).

*Act 50:*    think:  Now I can try to move an orange block on the # 1 base, a blue block on the # 2 base, and a yellow block on the # 3 base.

*Obs 51:*  OK.

*Act 52:*  move_to(0.71, 0.37, 0.37, -0.26)

*Obs 55:*     The # 1 base is in (0.37, -0.26).  The # 2 base is in (0.41, -0.28).  The # 3 base is in (0.45, -0.31).  A yellow block is in (0.41, -0.28).  A blue block is in (0.31, 0.34).  A orange block is in (0.37, -0.25).  A blue block is in (0.67, 0.19).  A blue block is in (0.45, -0.31).  A yellow block is in (0.71, 0.37).

*Act 56:*     think:  Now the orange block is on # 1 base, the yellow block is on # 2 base, and the blue block is on # 3 base, but the task has not finished yet.  So I might have placed the blocks in a wrong order.  We need to put different colors on different blocks, so let me move a yellow block on the # 1 base, a blue block on the # 2 base, and an orange block on the # 3 base.  But first, I need to remove the blocks from the bases.

*Obs 57:*  OK.

*Act 58:*  move_to(0.41, -0.28, 0.71, 0.37)

*Obs 61:*    The # 1 base is in (0.37, -0.26).  The # 2 base is in (0.41, -0.28).  The # 3 base is in (0.45, -0.31).  A blue block is in (0.31, 0.34).  A yellow block is in (0.71, 0.37).  A blue block is in (0.67, 0.19).  A yellow block is in (0.71, 0.37).  A orange block is in (0.67, 0.19).  A blue block is in (0.34, 0.15).

*Act 62:*     think:  Now I can try to move a yellow block on the # 1 base, a blue block on the # 2 base, and an orange block on the # 3 base.

*Obs 63:*  OK.

*Act 64:*  move_to(0.71, 0.37, 0.37, -0.26)

*Obs 67:*    The # 1 base is in (0.37, -0.26).  The # 2 base is in (0.41, -0.28).  The # 3 base is in (0.45, -0.31).  A blue block is in (0.45, -0.31).  A yellow block is in (0.71, 0.37).  A blue block is in (0.31, 0.34).  A yellow block is in (0.37, -0.25).  A orange block is in (0.41, -0.28).  A blue block is in (0.71, 0.2).

*Act 68:*     think:  Now the yellow block is on # 1 base, the orange block is on # 2 base, and the blue block is on # 3 base, but the task has not finished yet.  So I might have placed the blocks in a wrong order.  We need to put different colors on different blocks, so let me move a blue block on the # 1 base, an orange block on the # 2 base, and a yellow block on the # 3 base.  But first, I need to remove the blocks from the bases.

*Obs 69:*  OK.

*Act 70:*  move_to(0.45, -0.31, 0.31, 0.34)

*Obs 73:*     The # 1 base is in (0.37, -0.26).  The # 2 base is in (0.41, -0.28).  The # 3 base is in

(0.45, -0.31). A blue block is in (0.31, 0.34). A
orange block is in (0.71, 0.37). A yellow block is
in (0.74, 0.05). A blue block is in (0.69, 0.2). A
yellow block is in (0.71, 0.37). A blue block is in
(0.31, 0.34).
   *Act 74:* think: Now I can try to move a blue block
on the # 1 base, an orange block on the # 2 base, and
a yellow block on the # 3 base.
   *Obs 75:* OK.
   *Act 76:* move_to(0.31, 0.34, 0.37, -0.26)
   *Obs 79:* The # 1 base is in (0.37, -0.26). The
# 2 base is in (0.41, -0.28). The # 3 base is in
(0.45, -0.31). A yellow block is in (0.71, 0.37).
A blue block is in (0.45, -0.31). A blue block is
in (0.31, 0.34). A blue block is in (0.31, 0.34). A
orange block is in (0.41, -0.28). A yellow block is
in (0.18, -0.13).
   *Act 80:* think: Now the blue block is on # 1 base,
the orange block is on # 2 base, and the yellow block
is on # 3 base, but the task has not finished yet. So
I might have placed the blocks in a wrong order. We
need to put different colors on different blocks, so
let me move a yellow block on the # 1 base, a blue
block on the # 2 base, and an orange block on the # 3
base. But first, I need to remove the blocks from the
bases.
   *Obs 81:* OK.
   *Act 82:* move_to(0.45, -0.31, 0.74, 0.05)
   *Obs 85:* The # 1 base is in (0.37, -0.26). The
# 2 base is in (0.41, -0.28). The # 3 base is in
(0.45, -0.31). A blue block is in (0.31, 0.34). A
orange block is in (0.71, 0.37). A yellow block is
in (0.71, 0.37). A blue block is in (0.74, 0.05). A
blue block is in (0.31, 0.34). A yellow block is in
(0.18, -0.13).
   *Act 86:* think: Now I can try to move a yellow
block on the # 1 base, a blue block on the # 2 base,
and an orange block on the # 3 base.
   *Obs 87:* OK.

A successful example for ABA in task 2 is:

   *Obs 1:* The # 1 base is in (0.28, 0.21). The # 2
base is in (0.33, 0.22). The # 3 base is in (0.38,
0.23). The # 4 base is in (0.43, 0.23). A purple
block is in (0.35, 0.06). A blue block is in (0.46,
-0.06). A orange block is in (0.55, -0.15). A purple
block is in (0.43, -0.13). A green block is in (0.7,
0.3). A blue block is in (0.67, -0.01). You task is:
Place the blocks on the corresponding bases.
   *Act 2:* think: There are four bases and four
kinds of blocks. We need to put different colors on
different bases. Let me ask which color should be put
in the # 1 base.
   *Obs 3:* OK.
   *Act 4:* ask: Which color should be put on the # 1
base?
   *Obs 5:* You should put the purple block on the # 1
base.

```
    Act 6:   move_to(0.35, 0.06, 0.28, 0.21)
    Obs 7:     The # 1 base is in (0.28, 0.21).  The # 2
base is in (0.33, 0.22).  The # 3 base is in (0.38,
0.23).  The # 4 base is in (0.43, 0.23).  A blue block
is in (0.67, -0.01).  A orange block is in (0.55,
-0.15).  A purple block is in (0.28, 0.21).  A purple
block is in (0.42, -0.13).  A green block is in (0.7,
0.3).  A blue block is in (0.46, -0.06).
    Act 8:   think:  Now there is a purple block on the
# 1 base.  Let me ask which color should be put on the
# 2 base.
    Obs 9:   OK.
    Act 10:   ask:  Which color should be put on the # 2
base?
    Obs 11:   You should put the orange block on the # 2
base.
    Act 12:   move_to(0.55, -0.15, 0.33, 0.22)
    Obs 13:     The # 1 base is in (0.28, 0.21).  The # 2
base is in (0.33, 0.22).  The # 3 base is in (0.38,
0.23).  The # 4 base is in (0.43, 0.23).  A green
block is in (0.7, 0.3).  A orange block is in (0.33,
0.22).  A blue block is in (0.46, -0.06).  A purple
block is in (0.42, -0.13).  A blue block is in (0.67,
-0.01).  A purple block is in (0.28, 0.21).
    Act 14:   think:  Now there is a purple block on the
# 1 base and an orange block on the # 2 base.  Let me
ask which color should be put on the # 3 base.
    Obs 15:   OK.
    Act 16:   ask:  Which color should be put on the # 3
base?
    Obs 17:     You should put the blue block on the # 3
base.
    Act 18:     think:  Now I have put a purple block on
the # 1 base, an orange block on the # 2 base, and I
know that I should put a blue block on the # 3 base.
Since we need to put different colors on different
bases, I should put a green block on the # 4 base.
So, I should put a blue block on the # 3 base, and put
a green block on the # 4 base.
    Obs 19:   OK.
    Act 20:   move_to(0.46, -0.06, 0.38, 0.23)
    Obs 22:     The # 1 base is in (0.28, 0.21).  The #
2 base is in (0.33, 0.22).  The # 3 base is in (0.38,
0.23).  The # 4 base is in (0.43, 0.23).  A blue block
is in (0.67, -0.01).  A purple block is in (0.42,
-0.13).  A green block is in (0.43, 0.23).  A blue
block is in (0.38, 0.23).  A orange block is in (0.33,
0.22).  A purple block is in (0.28, 0.21).
```

## K   TRAJECTORIES OF THE REAL WORLD OPEN DOMAIN EMBODIED TASKS ENVIRONMENTS

Below is the prompt we use, this correspond to Figure 4 (a):

```
    Instruction and Observation 1:   I have just arrived
at the lab.  Can you help me to prepare for today's
work?  Observation:  a desk with a computer monitor,
keyboard and mouse Think 2:   To prepare for the work,
first, let's get the computer ready.  Let me make sure
```

whether the human needs to work with a computer or laptop.

*Observation 3:* OK.

*Ask 4:* Do you need to work with a computer or laptop? I can help you prepare with that.

*Observation 5:* Yes. Please help me connect the laptop to the monitor.

*Think 6:* Now I need to find a laptop, connect the laptop to the monitor, and turn on the laptop. But I don't know where is the laptop. Let me ask for that.

*Observation 7:* OK.

*Ask 8:* Can you tell me where is the laptop?

*Observation 9:* The laptop is on the desk next to me.

*Think 10:* Now I need to go to the next desk, take the laptop and bring it back. Then, I need to connect the laptop to the monitor.

*Observation 11:* OK.

*Action 12:* go to the next desk.

*Observation 13:* a laptop computer sitting on a desk in an office

*Action 14:* take the laptop.

*Observation 15:* you take the laptop.

*Action 16:* go to the original desk.

*Observation 17:* a desk with a computer monitor, keyboard and mouse

*Action 18:* connect the laptop to the monitor.

*Observation 19:* you connect the laptop to the monitor.

*Think 20:* Now I have finished the task. Then, let me ask what else does the human need.

*Observation 21:* OK.

*Ask 22:* Do you need anything else? I can help prepare them for you.

*Observation 23:* No, I only need to work with my laptop.

*Think 24:* To help create a better mood for work, I can bring the human something to drink. Let me ask for the preference.

*Observation 25:* OK.

*Ask 26:* Do you want something to drink? If so, what do you want?

*Observation 27:* Yes, please bring me something to drink from the pantry.

*Think 28:* Let me go to the pantry, take something to drink, and bring it back.

*Observation 29:* OK.

*Action 30:* go to the pantry.

*Observation 31:* a kitchen sink with two bottles sitting on it

*Action 32:* take a bottle of water.

*Observation 33:* you take a bottle of water.

*Action 34:* go back to the original table.

*Observation 35:* a desk with a computer monitor, laptop, keyboard and mouse

*Action 36:* put the bottle on the table

*Observation 37:* you put the bottle on the table

*Think 38:* Now I have helped prepare the computer and the drinks. Let me ask whether there is anything else I can do.

*Observation 39:* OK.
*Ask 40:* Is there anything else I can do for you?
*Observation 41:* No. Thank you.

Here is the evaluation result corresponds to Figure 4 (b):

*Instruction and Observation 1:* I'm leaving my office. Can you help me pack up? Observation: The image features a laptop computer placed on a desk or table, along with a water bottle and a backpack. There are two chairs in the scene, one closer to the laptop and the other further away. A mouse can also be seen on the desk, possibly connected to the laptop.
*Think 2:* To help pack up, first, let's disconnect the laptop and put it inside the backpack. Let me make sure whether the human needs to pack anything else.
*Observation 3:* OK.
*Ask 4:* Besides the laptop, do you need to pack anything else?
*Observation 5:* Yes, I need my mouse, and please fill up the water bottle.
*Think 6:* Now I need to disconnect the laptop, pack the laptop, mouse, and fill up the water bottle.
*Observation 7:* OK.
*Action 8:* disconnect the laptop.
*Observation 9:* You disconnect the laptop.
*Action 10:* put the laptop and mouse in the backpack.
*Observation 11:* You put the laptop and mouse in the backpack.
*Think 12:* Now I need to fill up the water bottle. Let me go to the kitchen sink and fill it up.
*Observation 13:* OK.
*Action 14:* go to the kitchen sink with the water bottle.
*Observation 15:* The image features a stainless steel kitchen sink placed in the center of the room, surrounded by various appliances and utensils.
*Action 16:* fill up the water bottle.
*Observation 17:* You fill up the water bottle.
*Action 18:* go back to the original table.
*Observation 19:* You go back to the original table.
*Action 20:* put the filled water bottle in the backpack.
*Observation 21:* You put the filled water bottle in the backpack.
*Think 22:* Now I have helped pack up the laptop, mouse, and filled water bottle. Let me ask whether there is anything else I can do.
*Observation 23:* OK.
*Ask 24:* Is there anything else I can do for you before you leave?
*Observation 25:* Oh, yes, please bring me the notebook.
*Think 26:* Now I need to find the notebook and bring it. Let me ask where the notebook is.
*Observation 27:* OK.
*Ask 28:* Can you tell me where the notebook is?
*Observation 29:* It's on the sofa.

*Think 30:*    Now I need to go to the sofa, take the notebook, and bring it back.
*Observation 31:* OK.
*Action 32:* go to the sofa.
*Observation 33:*    The image features a couch with a black book placed on top of one of the cushions. The couch is situated in the middle of the room and appears to be the focal point of the space.  In addition to the couch, there are several yellow and gray ottomans scattered around the room.  Some of these ottomans are located near the couch, while others are positioned closer to the edges of the room. Overall, the furniture creates a cozy and inviting atmosphere in the space.
*Action 34:* take the notebook.
*Observation 35:* You take the notebook.
*Action 36:* go back to the original table.
*Observation 37:* You go back to the original table.
*Action 38:* hand the notebook to the human.
*Observation 39:* Thank you.
*Ask 40:* Is there anything else I can do for you?
*Observation 41:* No.  Thank you.

Here is the evaluation result corresponds to Figure 4 (c):

*Instruction and Observation 1:*    Instruction:  I'm going swimming.  Can you help me prepare my things and put them in this bag?  Observation:  The image features a white mesh bag placed on a wooden floor. The bag has blue straps attached to it, which can be seen draped over the side of the bag. *Think 2:* To help the human prepare their things for swimming, I might need to gather items like a swimsuit, towel, goggles, swimming cap, and flip-flops.  Let me ask for their preferences.
*Observation 3:* OK.
*Ask 4:*    What items do you need for swimming?  For example, a swimsuit, towel, goggles, swimming cap, and flip-flops?
*Observation 5:* Yes, these are enough.
*Think 6:* Now I need to find these items and put them in the bag.  Let me ask where I can find these items.
*Observation 7:* OK.
*Ask 8:*        Can you tell me where I can find the swimsuit, towel, goggles, swimming cap, and flip-flops?
*Observation 9:*    The towel is already in the bag. The swimsuit and goggles are on the counter.  The flip-flops are next to the door.
*Think 10:*  First, I'll go to the counter to get the swimsuit and goggles.  Then, I'll go to the door to get the flip-flops.  Finally, I'll put everything in the bag.
*Observation 11:* OK.
*Action 12:* go to the counter.
*Observation 13:* The image features a laid-out outfit consisting of a pair of swim trunks and a pair of goggles.  The swim trunks are placed on the right side

of the image, while the goggles are positioned on the
left side.
   *Action 14:* take the swimsuit and goggles.
   *Observation 15:* You take the swimsuit and goggles.
   *Action 16:* go to the door.
   *Observation 17:*   The image features two pairs of
colorful flip-flops placed side by side on a wooden
floor.
   *Ask 18:*  Which pair of flip-flops would you like me
to take, the green and yellow ones or the blue and
white ones?
   *Observation 19:* The blue and white one.
   *Action 20:* take the blue and white flip-flops.
   *Observation 21:*    You take the blue and white
flip-flops.
   *Action 22:* go back to the bag.
   *Observation 23:* You go back to the bag.
   *Action 24*  put the swimsuit, goggles, and blue and
white flip-flops in the bag.
   *Observation 25:*  You put the swimsuit, goggles, and
blue and white flip-flops in the bag.
   *Think 26:*   Now I have helped prepare the items for
swimming. Let me ask whether there is anything else I
can do.
   *Observation 27:* OK.
   *Ask 28:* Is there anything else I can do for you?
   *Observation 29:* No. Thank you.

Here is the evaluation result corresponds to Figure 4 (d):

   *Instruction and Observation 1:*   Instruction: Now
I have ended a day's work. Can you help me relax?
Observation: The image features a desk with a laptop
and a book placed on it. The laptop is positioned
towards the left side of the desk, while the book is
closer to the right side. There are two clocks on
the desk, one located near the laptop and the other
closer to the center of the desk. A mouse can also
be seen on the desk, close to the laptop. In addition
to these objects, there is a pen in the scene as well.
*Think 2:*  To help the human relax, I can suggest some
activities. But first, let me ask for preferences to
better understand the needs.
   *Observation 3:* OK.
   *Ask 4:*   What kind of activities do you prefer to
relax? Do you want to read the book, listen to music,
or do something else?
   *Observation 5:*  Listen to music. Please also bring
me something to drink.
   *Think 6:*   I need to find a music player and bring
something to drink. Let me ask for their locations.
   *Observation 7:* OK.
   *Ask 8:*  Where can I find the music player and what
would you like to drink?
   *Observation 9:*    You can play the music via the
laptop. Do you have any suggestions for the drinks?
   *Think 10:* Let me suggest some options for drinks to
help the human decide.
   *Observation 11:* OK.

*Ask 12:* Do you prefer a hot or cold drink? I can suggest coffee, tea, or a cold beverage like water or juice.
*Observation 13:* Please bring me some tea.
*Think 14:* Now I need to go to the kitchen, prepare the tea, and bring it back. But first, let me play the music.
*Observation 15:* OK.
*Action 16:* play music on the laptop.
*Observation 17:* Please specify the music to play.
*Ask 18:* What type of music would you like me to play?
*Observation 19:* As long as it's relaxing, can you list some of them?
*Ask 20:* Sure, do you prefer classical, jazz, or ambient music?
*Observation 21:* jazz.
*Action 22:* play relaxing jazz music on the laptop.
*Observation 23:* you play relaxing jazz music on the laptop.
*Think 24* Now I need to go to the kitchen, prepare the tea, and bring it back.
*Observation 25:* OK.
*Action 26:* go to the kitchen.
*Observation 27:* Invalid action, there is no kitchen.
*Ask 28:* Where can I find the tea making supplies?
*Observation 29:* There is a bottle of tea on the shelf.
*Ask 29:* take the bottle of tea from the shelf.
*Observation 30:* The image features a wooden shelf with two bottles placed on it. One bottle is closer to the left side of the shelf, while the other is closer to the right side. Both bottles are positioned near the top of the shelf. In addition to the bottles, there is a teapot on the shelf, located between the two bottles. The teapot is positioned closer to the left side of the shelf.
*Action 31:* take the bottle of tea and the teapot.
*Observation 32:* You take the bottle of tea and the teapot.
*Ask 33:* go back to the original table.
*Observation 34:* You go back to the original table.
*Ask 34:* put the tea and teapot on the table.
*Observation 35:* You put the tea and teapot on the table.
*Think 36:* Now I have helped the human relax by playing music and bringing tea. Let me ask whether there is anything else I can do.
*Observation 37:* OK.
*Ask 38:* Is there anything else I can do for you?
*Observation 39:* No. Thank you.

## L  STATISTICS ON ASKING IN ALFWORLD

|  | # Questions |
|---|---|
| ALFWorld | 1.1 |
| ALFWorld with ambiguous tasks | 1.39 |
| Multiround ALFWorld | 2.53 |

Table 3: Statistics data on the average number of questions asked in successful episodes on all the 3 tasks reported in the paper with V7B model.

