# OpenReview forum: "Asking Before Acting: Gather Information in Embodied Decision-Making with Language Models"
_ICLR.cc/2024/Conference — Submitted to ICLR 2024_

### Official Review · Reviewer_umUa · 2023-10-28

**Soundness:** 3 good
**Presentation:** 2 fair
**Contribution:** 2 fair
**Rating:** 6
**Confidence:** 3

**Summary:**

The paper presents a theoretical framework, conceptualizing scenarios as a Markov Decision Process (MDP). This framework harnesses active querying to efficiently extract information from a language model. Remarkably, the proposed method can tweak its queries to be relevant even with slight alterations to existing agents, ensuring that previously acquired information is both retained and effectively repurposed. The examined issue is both relevant and significant. Agents equipped with the ability to tap into external knowledge repositories exhibit enhanced capability and safety over their counterparts. The integration of LLMs with assistance-seeking mechanisms is a novel endeavor. The results outperform the established baseline, and diverse experimental setups have been designed to underscore the method's efficacy.

**Strengths:**

-The paper presents a unique framework for assessing embodied decision-making, enabling agents to proactively seek information.
-Comprehensive tests were carried out on ALFWord and its derivatives, affirming the method's efficacy.
-The paper is well-organized and clear-presented.

**Weaknesses:**

- The writing could benefit from improvements, particularly typographical errors like "suboptimal beahviors" found in the second paragraph.
- Why choose to incorporate solely a human model rather than adopting approaches like the RLHF for human feedback?
- The authors' pursuit to tackle intricate issues in embodied AI, especially everyday tasks beyond mere embodied navigation, is praiseworthy. Nonetheless, showcasing the method's versatility across various realms, including embodied navigation, would enhance the paper's value.
- It would also be beneficial if the paper could cover works from pre-LLM era on embodied AI task that takes in help signal to guide its downstream task:
1.Chi, T.C., Shen, M., Eric, M., Kim, S. and Hakkani-tur, D., 2020, April. Just ask: An interactive learning framework for vision and language navigation. In Proceedings of the AAAI Conference on Artificial Intelligence (Vol. 34, No. 03, pp. 2459-2466).
2.Zhang, J., Yu, S., Duan, J. and Tan, C., 2022. Good Time to Ask: A Learning Framework for Asking for Help in Embodied Visual Navigation. arXiv preprint arXiv:2206.10606.
3.Singh, Kunal Pratap, Luca Weihs, Alvaro Herrasti, Jonghyun Choi, Aniruddha Kembhavi, and Roozbeh Mottaghi. "Ask4help: Learning to leverage an expert for embodied tasks." Advances in Neural Information Processing Systems 35 (2022): 16221-16232.

**Questions:**

All my questions are asked in the weakness section.

**Details Of Ethics Concerns:**

nill

---

> ### Author Response · Authors · 2023-11-16
> **Response**
>
> Thanks for your time and efforts in reviewing our paper! We highly appreciate your thoughtful and constructive suggestions. Your thoughtful and constructive suggestions have been invaluable to us, and we have carefully considered each comment. Our responses to your queries are outlined below:
>
> ---
>
> ***Q1. why not use RLHF?***
>
> In summary, we use human models during evaluation, to enable real time human interaction and save human efforts, while RLHF is a training method to learn human values. To be specific:
>
> - In ABA, we focus on the human (external information source) in the loop setting, where the agent can interact with humans in real time during deployment.
>
>     While typical RLHF methods like [1] collect human feedback dataset and train the model. During rollout, RLHF cannot help model the interactions with humans.
>
> - In ABA, for the same question, the human may provide different answers in different scenarios. For instance, asking "where is the cup?" may yield different responses in different rooms
>
>     While in RLHF, the learning objective is the concept of values / preferences, which is fixed, and cannot fit in our setting.
>
> While we have not incorporated RLHF in the current work, we acknowledge the potential for future research to explore combining RLHF to enhance the asking quality of ABA.
>
>
> [1] Ouyang, Long, et al. "Training language models to follow instructions with human feedback." Advances in Neural Information Processing Systems 35 (2022): 27730-27744.
>
> ---
>
> ***Q2,4. navigation and missing related works***
>
> Thanks for pointing out these related works! We have added them to the related work section and highlighted the modifications in blue.
>
> ---
>
> ***Q3. writing issues***
>
> Thanks! We have fixed the typos in our manuscript and these modifications are highlighted in blue for your convenience.
>
> ---
>
> We sincerely appreciate your thorough review and constructive suggestions. If you have any further comments or inquiries, please do not hesitate to let us know. And we genuinely hope for your reconsideration of the score.

---

> > ### Comment · Reviewer_umUa · 2023-11-22
> > **Response**
> >
> > Thanks to the author for addressing all my doubts, however after much consideration, i would still like to keep my rating.

---

> ### Author Response · Authors · 2023-11-21
>
> Dear Reviewer, we would like to thank you again for your efforts and time in providing thoughtful feedback and comments. We’ve revised the paper according to your suggestions and replied to all the questions and concerns. Since the discussion period is ending soon, we would greatly appreciate it if you could let us know whether you have any additional comments.
>
> Thanks a lot!

---

### Official Review · Reviewer_mCAD · 2023-10-30

**Soundness:** 2 fair
**Presentation:** 2 fair
**Contribution:** 2 fair
**Rating:** 3
**Confidence:** 4

**Summary:**

In this paper, a new prompting and LM fine-tuning method is introduced. The goal of Asking Before Acting (ABA) is to gather information (“asking”) from the environment or external sources before performing an action (“acting”). In addition to a zero-shot method, there is also ABA-FT which is fine-tuned on labeled trajectories.

**Strengths:**

- Originality: The paper addresses an inefficiency of current LLMs in embodied LM. It is much easier to directly ask a question than directly explore the environment. This is also an intuitive idea.
- Significance: ABA could provide shorter policies. The new versions of AlfWorld provide a more challenging version of a widely used benchmark.

**Weaknesses:**

- Comparison with previous works: One of the major weaknesses is that several works with competitive baselines are omitted from the paper. For example, in AlfWorld, Reflexion (1) and AdaPlanner (2) achieve similar or better results on AlfWorld than ABA. For the robotics task, there is no comparison with works like Cliport (3). For the finetuned version of ABA, there is no comparision with finetuned models or imitation learning methods.
- For the new dataset variants introduced, Multiround AlfWorld and Ambigious AlfWorld are not evaluated on methods other than ReACT, ABA, ABA-FT.
- A key part of ABA is receiving/guidance from the environment or external sources. However, in the paper, a second LLM is used as the 'human' or external source. There is no ablation showing that the external source is providing guidance and not just the answer.
- An evaluation on more decision making datasets such as programming datasets, block world, etc. would strengthen the paper significantly given the general high performance of LLMs on AlfWorld.

(1): Shinn, Noah, Beck Labash, and Ashwin Gopinath. "Reflexion: an autonomous agent with dynamic memory and self-reflection." arXiv preprint arXiv:2303.11366 (2023).
(2): Haotian Sun, Yuchen Zhuang, Lingkai Kong, Bo Dai, & Chao Zhang. (2023). AdaPlanner: Adaptive Planning from Feedback with Language Models.
(3): Shridhar, Mohit, Lucas Manuelli, and Dieter Fox. "Cliport: What and where pathways for robotic manipulation." Conference on Robot Learning. PMLR, 2022.

**Questions:**

- What is the expert policy used to train ABA-FT?
- Is there any visual model used for the robotic control experiments?

---

> ### Author Response · Authors · 2023-11-16
> **Response**
>
> Thanks for your time and efforts in reviewing our paper! We highly appreciate your thoughtful and constructive suggestions. We have carefully considered each comment. Our responses to your queries are outlined below:
>
> ---
>
> ***Q1, Q2 and Q4. insufficient experiments, new baselines and benchmarks***
>
> **New baselines:**
>
> We've carefully considered your recommendations, and here's our feedback:
>
> 1. *Reflexion [1] and Adaplanner [2]:*
>
>     These methods focus on learning from previous mistakes through multiple rollouts, while ABA aims to perform tasks within a single shot. Therefore we are orthogonal and focus on different settings. We acknowledge the potential for future research to explore combinations with [1,2] to enhance performance.
>
>
> 2. *CLIPort [3]*
>
>    We believe [3] is not suitable here since:
>
>    - Different settings. Both ABA and ReAct are few-shot methods, while [3] needs to collect an expert dataset to do imitation learning.
>
>    - In section 4.2, our tasks involve the agent asking questions or reasoning to gather information, a scenario not directly adaptable to imitation learning-based methods like [3].
>
> 3. *BUTLER in ALFWorld variants*
>
>     We have already compared BUTLER in the original ALFWorld. We show ABA which use $K=2$ trajectories can significantly outperform BUTLER with $10^5$ training trajectories. For similar but more challenging environments, we believe it is not necessary to compare with it again.
>
>
> **New benchmarks:**
>
> Regarding new environments, our experiments cover three types, including text-based tasks, robot arm manipulation, and real-world embodied environments, totaling 29 different tasks. As acknowledged by other reviewers, our experiments are “thorough” (from P7Hr), “sufficient” (from 23Y9), and “comprehensive” (from umUa) experiments.
> If there are specific aspects or types of experiments you believe are lacking or could be supplemented, we would greatly appreciate further clarification and any suggested benchmarks.
>
> ---
>
>
> ***Q3.  There is no ablation showing that the external source is providing guidance and not just the answer.***
>
> In Section 3.3 and Appendix A, we showed how the human model is designed. It prompts the LLM with environmental information and is prompted to answer the questions. The human model does not have access to the oracle action, and it will not provide the answer.
>
> ---
>
> ***Q5. What is the expert policy used to train ABA-FT?***
>
> As outlined in Appendix E, we designed a rule-based policy according to the PDDL planning trajectories provided along with the environment to collect the data.
>
> ---
>
> ***Q6. Is there any visual model used for the robotic control experiments?***
>
> As outlined in appendix J, we use a hand designed captioning module, which can access the simulator information and label the object type and its ground truth coordinates. As discussed in Huang et al. (2023), this module can be implemented through the depth image and a combination of object detection model, an object segmentation model, and a video tracker. But we temporarily leave this module hand-designed since it is not the key point of our work.
>
> ---
>
> We appreciate your thorough evaluation and remain open to further discussion or clarification on any points. And we genuinely hope for your reconsideration of the score.

---

> ### Author Response · Authors · 2023-11-21
>
> Dear Reviewer, we would like to thank you again for your efforts and time in providing thoughtful feedback and comments. We’ve revised the paper according to your suggestions and replied to all the questions and concerns. Since the discussion period is ending soon, we would greatly appreciate it if you could let us know whether you have any additional comments.
>
> Thanks a lot!

---

> > ### Comment · Reviewer_mCAD · 2023-11-21
> >
> > Thank you to authors for addressing my concerns. However, I still keep my score at a 3. While formally, the setting of ABA (not finetuned) is different than Reflexion and Adaplanner, in practice it does not seem so different. In terms of using an LLM as the human and the datasets used as evaluation, prompting does not eliminate the potential for simply providing the answer or for being incorrect. Additionally, I agree with reviewer  P7Hr's review.

---

> > > ### Author Response · Authors · 2023-11-22
> > >
> > > Thanks for your feedback.
> > >
> > > > While formally, the setting of ABA (not finetuned) is different than Reflexion and Adaplanner, in practice it does not seem so different.
> > >
> > > Thank you for acknowledging that “the setting of ABA (not finetuned) is different from Reflexion formally”. However, as for the claim “in practice it does not seem so different”, we respectfully disagree.
> > >
> > > In our previous response, we have pointed out that Reflexion [1] and Adaplanner [2] allow rollout in the same environment for multiple trials and focus on learning from previous mistakes. After each trial, they evaluate the trajectory via some evaluator $M_e$ and do “reflection” to improve the policy via LLM reasoning. Finally, they report performance after several such trials. In contrast to this few-shot setting, our method focuses on solving the task within 1 trial by asking open-ended questions to gather information.
> > >
> > > In [1] Fig 3 (a), they report the successful rate of Reflexion. The x-axis is the number of trials. When $x=0$, where only 1 trial is allowed, which is the same setting as ours, Reflexion deteriorates to ReAct. We already incorporate ReAct as our baseline. If the reviewer believes whether allowing multiple trials is not important and are the same in practice, then Reflexion will equal to ReAct, which, obviously, is not the case.
> > >
> > > ---
> > >
> > > > In terms of using an LLM as the human and the datasets used as evaluation, prompting does not eliminate the potential for simply providing the answer or for being incorrect.
> > >
> > >
> > > We believe this can be divided into two questions:
> > >
> > > **Q1** Will the human model provide the answer, i.e., the ground truth actions, to the agent given prompting?
> > >
> > > This question looks the same as previous W3. In our previous rebuttal, we have explained how the model is designed, how it is prompted, and we have pointed out the fact that the human model does not have access to the ground truth action, so it will not provide this to the agent.
> > >
> > > To avoid any potential misunderstanding, we would like to further provide an example. As shown in Figure 3, the action space of the agent is the $(x_1, y_1, x_2, y_2)$, which specifies the pick and place coordinates. However, the human model only knows about the preference over the cubes (e.g., there are five cubes and the second one from left if preferred) and the color correspondence of placement (e.g., the first place matches the red color, the second matches the green color), and it does not know the task, the coordinates, or the ground truth action. Thus, it will not and can not answer the ground truth action since it does not know the answer.
> > >
> > > **Q2** Will the human model be incorrect sometimes?
> > >
> > > We thank the reviewer for raising this new question. However, this has already been discussed in our original manuscript. As outlined in Section 4.1 and Appendix C, we carried out an ablation study and showed analysis to this question.

---

> > > > ### Comment · Reviewer_mCAD · 2023-11-22
> > > >
> > > > Thank you for the clarifications from the authors.
> > > > - As for the comparison between ABA and previous works, I do acknowledge that ABA does not use an evaluator as part of the learning process and can be considered zero-shot. That being said, ABA still requires multiple 'asking' interactions or calls to the 'human' before an answer is give (as noted in section 3.2, second paragraph in the paper: "By providing K such trajectories, we learn through ICL .."). Previous works and ABA both use some form of prompting with environmental feedback. While ABA is different, previous works significantly limit novelty.
> > > > - Use of LLM as human in terms of oracle answer. Thank you for the authors for the example. I agree that for the robot arm manipulation task this not an a problem. However, this is not necessarily true for AlfWorld. For the example given in Appendix B at the bottom of page 15, the correct action is 'put mug 1 in/on side table 1.' The LLM does have access to this information and can output the 'answer' instead of just the context to the answer.
> > > > - Correctness of LLM. Appendix C shows a large variance of the human model in terms of correct answers. No specific actions are taken to address hallucinations in ABA even though in the qualitative examples, hallucinations are pointed out in the baseline examples.

---

> ### Author Response · Authors · 2023-11-23
>
> Thanks for your feedback.
>
> ---
>
> **Q1.** Whether [1,2] should be used as baselines?
>
> We would like to begin by summarizing the overall discussion.
>
> - In the reviewer’s original review, mCAD asks for more baselines like [1-2].
>
> - In our original response, we explained these methods are not suitable for baselines since they have different settings. [1,2] allow rollout in the same environment multiple times, whenever they fail, they can restart the environment and learn from previous mistakes. In contrast, we focus on solving the task within 1 trial. Therefore, it is not proper to compare with [1,2] directly.
>
> - In the reviewer’s following feedback, mCAD agrees [1,2] and our method are different formally, but mCAD suggests they are similar in practice.
>
> - In our following response, we explained that it does not make sense to claim these few-shot methods incorporating environment restarting equals one-shot methods. We cite [1] to illustrate that with only 1 shot, [1] degenerate to ReAct, which is a baseline cited in our paper. So, [1,2] are not similar to our method in practice.
>
> - In the previous feedback, mCAD acknowledged that our method is zero-shot, while [1,2] are not, but mCAD proposed new similarities between our methods and [1,2]: (1) both methods need multiple interactions to give an answer, (2) both methods use prompting.
>
> This time, our response is as follows:
>
> First, we believe it is way enough to prove [1,2] cannot be directly used as baselines according to previous rebuttal.
>
> As for the new points raised by the reviewer, we believe mCAD might have misread our paper: reviewer mCAD seems to confuse the following three concepts (a) multiple actions are taken in 1 trajectory; (b) multiple offline trajectories used for In-Context-Learning as in ABA; (c) [1,2] include online trajectories as prompting.
>
> - In the reviewer’s feedback, mCAD claims we need to do several asking actions in 1 trajectory (type (a)), but mCAD cited the ICL part (type(b)) to support this claim.
>
> - mCAD suggested that we use ICL prompt (type (b)), while [1,2] also use prompting (which is type(c), but mCAD mistaking it for type(b)).
>
> Besides this misreading, it is quite weird to ask someone to compare two methods simply because both of these methods use prompting regardless of other settings.
>
> ---
>
> **Q2.** Whether the human model can output ground truth action?
>
> We thank the reviewer for acknowledging that the previous criticism does not exist in Section 4.2. Now let’s look at Section 4.1, ALFWorld, to solve the reviewer’s concern. To finish the task, the agent needs to find the object, pick it up, wash it / heat it / cool it, or even open a desk lamp to check it, before placing it to the target location. As outlined in Appendix A, the human model only knows the placement of all the objects. In this paper, we propose that, instead of finding the object by onerous trial and error, it is more efficient to query in natural language. Therefore, in the first step “finding the object”, the information may seem like the action, but this is exactly what the agent learned to ask. Furthermore, we would like to remind the reviewer that, for all the following steps, the human model does not know what task is assigned to the agent, it does not know what to do with the target object, it does not know the target location, so it can not produce the ground truth action.
>
> ---
>
> **Q3.** Should we address hallucination in our paper?
>
> We thank the reviewer for reading the related paragraph. However, as you may agree, the human model does not belong to the ABA method. It is just a proxy of real humans during evaluation, so as to evaluate ABA with minimal human involvement, which is not the key component of this paper. What's more, this result can be easily improved by using larger and stronger LLM. The Appendix C shows Vicuna 7B human model results, which only shows up in a small fraction of experiments in Section 4.1, while other results in Section 4.1 and Section 4.2 use GPT-35, GPT-4 as human model.
>
> The question about hallucination is raised out of nowhere. We do not even mention the word “hallucination” in any related analysis or in our main paper. Although LLMs are well known for suffering from hallucination, how to solve this problem, however, is out of the scope of this paper.

---

### Official Review · Reviewer_23Y9 · 2023-10-31

**Soundness:** 4 excellent
**Presentation:** 4 excellent
**Contribution:** 3 good
**Rating:** 6
**Confidence:** 5

**Summary:**

In this paper, the authors propose the ABA method to empower agents to gather external contextual information before decision-making, which is inspired by the behavior of humans completing tasks in unknown environments. Agents benefit from the newly proposed ABA paradigm and can avoid unnecessary trial and error, enhancing its efficiency and performance. Experiments demonstrate the effectiveness of the proposed methods.

**Strengths:**

- The proposed ABA idea is interesting. Compared to traditional self-explore agents, ABA agents can circumvent potentially laborious trial and error.
- Leveraging natural language and ICL to inquire information from humans or LLM is reasonable.
- The authors formulaically define the Contextual MDP with Human / External information sources in the loop based on the Contextual MDP, which provides a solid foundation to follow for future research.
- Extensive details and demonstrations are provided in the appendix, which makes the work easy to follow.
- The experiments are sufficient.

**Weaknesses:**

Weaknesses:
1. For embodied decision-making tasks, some related works need to be discussed. For example, previous works (e.g. [1]) explore the use of knowledge graph as external commonsense and extract pertinent information via GNN. Also, [2] extracts knowledge from VLM during the decision process. Both works aim to gather external information for improving navigation decisions in unseen environments, which is similar to the ABA idea.

    [1] Room-and-object aware knowledge reasoning for remote embodied referring expression.

    [2] Room-Object Entity Prompting and Reasoning for Embodied Referring Expression.

2. Given that the core of the ABA method is to obtain effective external assistance, and obtaining external assistance has costs such as time, resources, and human efforts. Therefore, how to balance the inquiry frequency and performance is a missing topic in this work.

3. There are still some typos, and the authors need to further polish the writing.
- Page 4: Missing comma in the sixth line of subsection 3.2.1
- Page 9: "open do- main image" in the conclusion section
- Page 14: "Section ??"

4. Combining the method with a real robot and verifying this work would be even better.

5. Suggestion: Further simplify the formula.

**Questions:**

See Weaknesses

---

> ### Author Response · Authors · 2023-11-16
> **Response**
>
> Thanks for your time and efforts in reviewing our paper! We highly appreciate your thoughtful and constructive suggestions. Your thoughtful and constructive suggestions have been invaluable to us, and we have carefully considered each comment. Our responses to your queries are outlined below:
>
> ---
>
> ***Q1. balance the performance and the efficiency***
>
> We appreciate your observation regarding the balance between performance and efficiency. In ABA, our key design focuses on prompting the agent to ask questions only when necessary, achieved through the injection of a proper prior. In the ABA-FT variant, we further enhance this efficiency by treating known information as a type of metadata, aiding the decision-making process on whether to ask questions.
>
> To demonstrate the efficiency of these designs, we conducted an ablation study by counting the total number of questions asked in V7B ALFWorld successful trials, and the results are as follows (which is also added to Appendix L):
>
> |                               | # Questions |
> |-------------------------------|-------------|
> | ALFWorld                      | $1.1$       |
> | ALFWorld with ambiguous tasks | $1.39$      |
> | Multiround ALFWorld           | $2.53$      |
>
> The results indicate that the average number of questions closely aligns with the minimum necessary questions. This study substantiates our claim that ABA effectively minimizes the number of questions asked. We will leave minimizing the number of questions to our future work.
>
> ---
>
> ***Q2. Fix the typos***
>
> Thanks! We have promptly rectified these errors in our manuscript, and the modifications have been highlighted in blue for your convenience.
>
> ---
>
> ***Q3. Missing related works***
>
> We are grateful for your recommendation of additional relevant works. We have incorporated these insightful contributions into our related works section, alongside other vision-language navigation works like TD-STP, SEvol which we believe to be related to the embodied decision making setting. The modifications have been highlighted in blue.
>
> ---
>
> We sincerely appreciate your thorough review and constructive suggestions. If you have any further comments or inquiries, please do not hesitate to let us know. And we genuinely hope for your reconsideration of the score.

---

> ### Author Response · Authors · 2023-11-21
>
> Dear Reviewer, we would like to thank you again for your efforts and time in providing thoughtful feedback and comments. We’ve revised the paper according to your suggestions and replied to all the questions and concerns. Since the discussion period is ending soon, we would greatly appreciate it if you could let us know whether you have any additional comments.
>
> Thanks a lot!

---

### Official Review · Reviewer_P7Hr · 2023-10-31

**Soundness:** 3 good
**Presentation:** 3 good
**Contribution:** 1 poor
**Rating:** 3
**Confidence:** 3

**Summary:**

This paper introduces a novel approach, "Asking Before Acting" (ABA), aimed at enhancing the decision-making efficiency of Large Language Models (LLMs) in unfamiliar environments. The core concept is inspired by human behavior, where individuals often seek additional information before taking action, thereby avoiding unnecessary trial and error. The ABA approach empowers agents to inquire proactively using natural language, enhancing their interaction within various environments. The paper begins by acknowledging the proficiency of LLMs in various tasks but highlights their inefficiency in environments with limited or ambiguous information. The ABA methodology is introduced as a solution, allowing agents to ask open-ended questions to gather essential information, leading to more efficient and informed decision-making. This approach is distinct from previous works that often restricted interactions or required human intervention for providing information. To further enhance ABA's performance, the authors introduce ABA-FT, which reformulates metadata associated with question formulation, helping the model understand the rationale behind asking questions. This fine-tuning process leads to notable improvements, especially in challenging tasks.

**Strengths:**

- The introduction of the ABA methodology is a novel concept in the realm of AI and decision-making. By enabling LLMs to ask questions before taking actions, the authors bridge the gap between human interactive learning and machine autonomy. This approach mimics human behavior, representing a shift towards more intuitive, adaptive AI systems.
- The authors have conducted thorough experiments across diverse environments, which adds credibility to their claims. By testing the ABA methodology in various scenarios, including text-based tasks, robot arm manipulations, and real-world open-domain tasks with image inputs, they demonstrate the model's versatility and applicability in different contexts.

**Weaknesses:**

- The problem in the paper is formed as Contextual MDP, while I think it's better formulating the problem into a more suitable domain like POMDP, which is much well-accepted and clearly defined. Is there a necessity to use Contextual MDP? I hope the authors can further explain the motivation.
- The paper's methodology heavily relies on the capabilities of Large Language Models (LLMs). I think the method itself does not make much sense. From my point of view, it seems just like the authors design some tasks with ambiguity and use the "ask before action" paradigm to help solve such ambiguity. But the model does not resolve such ambiguity by itself, i.e., through rapid trial-and-error. On the other hand, it still depends on the human to help it. So why not the human provide the complete information at the beginning? Besides, I don't see the proposed "ask before action" paradigm as a big contribution, as we have already see many works related to the LLM agency in which LLMs are driven to finish more complex interactions and tasks.
- There lacks error analysis conducted on instances where ABA underperformed or failed. What insights were derived from these analyses?  Besides, I think additional QA in ABA actually imports more information compared with other baselines, which definitely lead to better performance. Maybe other baselines should also use information like “the second red block from the left” for a equal comparison.

**Questions:**

See weakness above.

---

> ### Author Response · Authors · 2023-11-16
> **Response (2/2)**
>
> ***Q3. Why use Contextual MDP rather than POMDP:***
>
> Although Contextual MDP is a type of POMDP, we opted for the former because:
>
> - Contextual MDP provides a latent context $c$ which helps to model the task information, while POMDP does not.
>
> - $c$ helps better formulation: we can ask pertinent questions to infer the context $c$, the human model inside the environment $H$ will provide different answers to the same questions according to the different $c$. while POMDP can not.
>
> - Contextual MDPs are well defined and widely accepted, especially in RL generalization theory, multi-task, and meta RL domains. Please refer to [9] for a thorough survey.
>
> ---
>
> ***Q4. Error analysis on ABA:***
>
> We provided error analysis of a typical failure case in Appendix F, and we appended several new failure cases at the end of Appendix F (modifications are highlighted in blue). The analysis can be summarized as follows:
>
> Two key failure types are identified:
>
> - *undesirable answers from human models during evaluation*
>
>     Note we use human models to mimic humans during evaluation to minimize human involvement, but this may sometimes provide undesirable answers.
>
> - *mismatches between agent commonsense and the environment*
>
>     For instance, the task is to collect eggs from the garbage can, but the agent believes the eggs are no longer suitable for eating and proposes to buy new ones.
>
> We believe incorporating real humans during deployment and refining prompting can mitigate these failures, leaving room for future research. For detailed information, please refer to Appendix F.
>
> ---
>
> Once again, we sincerely appreciate your feedback, and we hope our responses address any concerns you may have. Your insights are crucial to the improvement of our work, and we welcome any additional feedback, and we genuinely hope for your reconsideration of the score.
>
> ---
>
> [1] Khanh Nguyen and Hal Daume ́ III. Help, anna! visual navigation with natural multimodal assistance via retrospective curiosity-encouraging imitation learning. arXiv preprint arXiv:1909.01871, 2019.
>
> [2] Khanh Nguyen, Debadeepta Dey, Chris Brockett, and Bill Dolan. Vision-based navigation with language-based assistance via imitation learning with indirect intervention. In Proceedings of the IEEE/CVF Conference on Computer Vision and Pattern Recognition, pp. 12527–12537, 2019.
>
> [3] Kunal Pratap Singh, Luca Weihs, Alvaro Herrasti, Jonghyun Choi, Aniruddha Kembhavi, and Roozbeh Mottaghi. Ask4help: Learning to leverage an expert for embodied tasks. Advances in Neural Information Processing Systems, 35:16221–16232, 2022b.
>
> [4] Felipe Leno Da Silva, Pablo Hernandez-Leal, Bilal Kartal, and Matthew E Taylor. Uncertainty- aware action advising for deep reinforcement learning agents. In Proceedings of the AAAI con- ference on artificial intelligence, volume 34, pp. 5792–5799, 2020.
>
> [5] Iou-Jen Liu, Xingdi Yuan, Marc-Alexandre Coˆte ́, Pierre-Yves Oudeyer, and Alexander Schwing. Asking for knowledge (afk): Training rl agents to query external knowledge using language. In International Conference on Machine Learning, pp. 14073–14093. PMLR, 2022.
>
> [6] Wenlong Huang, Fei Xia, Ted Xiao, Harris Chan, Jacky Liang, Pete Florence, Andy Zeng, Jonathan Tompson, Igor Mordatch, Yevgen Chebotar, et al. Inner monologue: Embodied reasoning through planning with language models. arXiv preprint arXiv:2207.05608, 2022b.
>
> [7] Shah, Dhruv, Błażej Osiński, and Sergey Levine. "Lm-nav: Robotic navigation with large pre-trained models of language, vision, and action." Conference on Robot Learning. PMLR, 2023.
>
> [8] Huang, Wenlong, et al. "Voxposer: Composable 3d value maps for robotic manipulation with language models." arXiv preprint arXiv:2307.05973 (2023).
>
> [9] Kirk, Robert, et al. "A survey of generalisation in deep reinforcement learning." arXiv e-prints (2021): arXiv-2111.

---

> ### Author Response · Authors · 2023-11-16
> **Response (1/2)**
>
> Thanks for your time and efforts in reviewing our paper! We highly appreciate your thoughtful and constructive suggestions. Your thoughtful and constructive suggestions have been invaluable to us, and we have carefully considered each comment. Our responses to your queries are outlined below:
>
> ---
>
> ***Q1. about the setting and contribution: the "asking before action" paradigm***
>
> After reading the review, we would like to clarify our setting in case of any potential misunderstandings:
>
> Currently, LLM based agents are designed to solve tasks on their own without external interactions, which is identified to be inefficient in scenarios with limited information.
> To address this concern, we concentrate on a special type of setting with human / external information sources in the loop (refer to Def 3.1), and propose the “asking before action” paradigm. This distinguishes our work from purely autonomous agents.
>
> We believe human / external information in the loop is important and promising [1-6].
> We are not primarily concerned with completing complex tasks through trial-and-error but with facilitating decision-making through interaction with humans.
>
> We take a significant step forward in the Human-in-the-Loop (HITL) setting. Previous works including [1-6] ask for oracle action, template questions, etc. ABA proposes gathering necessary information in natural language. This contributes to a novel setting, a set of newly designed benchmarks, and helps extend the boundary of current research.
>
> Looking ahead, we foresee combining ABA with recent advances in purely autonomous works, such as [7] for navigation and [8] for pick-and-place tasks, to build more intelligent embodied agents.
>
> **To summary, our contributions are:**
>
> 1. identify a novel setting (~Def 3.1) that can possibly contribute to building a more embodied AI in the real world.
>
> 2. design new methods which can mimic human behavior, and "representing a shift towards more intuitive, adaptive AI systems".
>
> 3. a set of benchmarks with a set of human model designs that can facilitate later research in this direction.
>
> ---
>
> ***Q2. about the scenarios: Why Not Provide Complete Information at the Beginning/Feed to Other Baselines?***
>
> We appreciate your question about complete information. We believe providing complete information is impractical for real-world tasks due to scale differences between simulators and reality.
>
> While simulators may handle scenarios with dozens of objects, the real world poses challenges in enumerating all relevant information. To illustrate, consider seeking assistance in preparing lunch—an exhaustive list of utensils and ingredients would require significant human effort and might overlook unforeseen circumstances. Therefore, for real-world scenarios, providing complete information is not viable, and we have not supplied it to other baselines.

---

> ### Author Response · Authors · 2023-11-21
>
> Dear Reviewer, we would like to thank you again for your efforts and time in providing thoughtful feedback and comments. We’ve revised the paper according to your suggestions and replied to all the questions and concerns. Since the discussion period is ending soon, we would greatly appreciate it if you could let us know whether you have any additional comments.
>
> Thanks a lot!

---

> > ### Comment · Reviewer_P7Hr · 2023-11-22
> >
> > Thank you very much for your response. I have carefully reviewed the rebuttals and the additional error analysis and some of them have addressed my concerns. However, there are still some reasons why I maintain my original stance: Q1: I do not doubt that the ABA paradigm can aid many tasks and contribute to building more intelligent agents. However, the idea seems simplistic and does not significantly advance beyond similar works. Q2: My issue is not related to the real-world applicability but concerns the fairness in how you compare your model with other baselines. Q3: Based on your explanation, I think it's reasonable to frame the problem as a Contextual MDP (but I think it is not rigorous to draw the conclusion "well defined and widely accepted" from the survey paper from arxiv). Q4: Thank you very much for your thorough error analysis.

---

> > > ### Author Response · Authors · 2023-11-22
> > >
> > > Thanks for your response.
> > >
> > > >  I do not doubt that the ABA paradigm can aid many tasks and contribute to building more intelligent agents. However, the idea seems simplistic and does not significantly advance beyond similar works.
> > >
> > > We express our gratitude for acknowledging the importance of the ABA paradigm in aiding diverse tasks and contributing to the development of more intelligent agents. This recognition underscores the significance of the setting itself.
> > >
> > > In our exploration of related works, it becomes evident that existing approaches fall short of achieving the generality present in our method. Unlike previous works with human-in-the-loop often seeking help for oracle actions or asking templated questions. In contrast, our approach stands out for its ability to query in natural language, marking a step forward in this important setting.
> > >
> > > Notably, our method is not simplistic. Can we further ask why the ideas seem simplistic? For our method, we incorporate in-context learning and fine-tuning to learn the asking rationales. If the “simplistic” means combining or prompting pre-trained LLM without, we would like to argue that like [1-3], it is common to combine method or pipeline designs with existing LLMs. In addition to the method part, we also develop a theoretical framework and the evaluation pipeline which minimizes human enrollment by a human model, and design several new test benchmarks for further studies in this domain.
> > >
> > > What's more, the empirical results further validate our assertions, demonstrating a substantial performance improvement over existing methods.
> > >
> > >
> > > [1] ReAct: Synergizing Reasoning and Acting in Language Models, ICLR 2023.
> > >
> > > [2] Language Models Meet World Models: Embodied Experiences Enhance Language Models, NeurIPS 2023
> > >
> > > [3] Leveraging Pre-trained Large Language Models to Construct and Utilize World Models for Model-based Task Planning, NeurIPS 2023
> > >
> > > ---
> > >
> > > > My issue is not related to the real-world applicability but concerns the fairness in how you compare your model with other baselines.
> > >
> > > First, In our previous response, as you may agree, we explained that it is not practical to prepare the complete information for the agent about every aspect of the world.
> > >
> > > What’s more, the world itself is partially observable for the agent. The agent itself needs to explore and gather the information. Previous work gathers the information through trial and error, while our work gathers the information via identifying the proper question and asking in natural language. This is the key contribution of our work. By preparing the complete information to the baselines (assume it is possible), we are actually making the environment fully observable, while our agent still operates in a POMDP. Maybe it is a good idea to provide the complete information as an oracle agent, and see the gap between it.

---

### Author Response · Authors · 2023-11-16
**Overall Response**

**Summary**

We sincerely appreciate all reviewers’ and ACs’ time and efforts in reviewing our paper. We thank you all for the insightful and constructive suggestions.

Regarding our paper's setting, we are thankful for the positive feedback, with Reviewer P7Hr deeming it "novel," 23Y9 finding it "interesting and reasonable," mCAD describing it as "intuitive," and umUa recognizing its "uniqueness".

Special acknowledgment goes to 23Y9, who notes that our theoretical modeling "provides a solid foundation for future research," P7Hr, highlighting that our method "bridges the gap between human interactive learning and machine autonomy," and umUa, commending our paper as "well-organized and clear-presented."

The experiments received favorable comments from P7Hr, 23Y9, and umUa, who found them "thorough," "sufficient and extensive," and "comprehensive," respectively.

---

**Here is a summary of our updates:**

- Additional error analysis (Appendix F) (P7Hr)

- Inclusion of missing related works (Section 5) (23Y9, mCAD, umUa)

- Adding ablation study on asking frequency (Appendix L) (23Y9)

- Correction of typos (General feedback)

We highlight all the modifications in blue in our manuscript.

---

**To further engage with the reviewers' feedback, we have initiated discussions on:**


- Further clarification on our setting and contribution (P7Hr)

- Discussion on Contextual MDP over POMDP (P7Hr)

- Discussion on complete information (P7Hr)

- Discussion on new baselines and benchmark (mCAD)

- Discussion on human model (mCAD)

- Discussion on comparison between human model with RLHF (umUa)

---

We owe many thanks to all reviewers’ extremely helpful suggestions. We hope we have addressed all of your concerns and will continue to polish our paper in our revision. Thanks again for your comments, and we sincerely wish you could reconsider your score.

---

### Meta-Review · Area_Chair_Rfe8 · 2023-12-05

**Metareview:**

The authors provide an implementation of the human-like ability to ask clarifications before executing embodied tasks.
Simple idea (explore in ICL and FT) and across diverse environments, showing interesting improvements. While including a way for the model to ask clarification is interesting, the use cases are a bit narrow (slightly contrived use cases) and the way the model is encouraged to ask questions (by generating a few handcrafted ICL examples or formulaic interactions to FT on) highlight that the task being learned is perhaps not that complex.

I'm on the fence but leaning towards it not passing the bar for acceptance at ICLR this year -- generalization to more general settings, or e.g. showing that there is interesting emergence of a model realizing that it doesn't have the information necessary for the problem, or further analyses of whether all the questions it asks are necessary / optimal would be great improvements.

**Justification For Why Not Higher Score:**

use cases are a bit narrow (slightly contrived use cases) and the way the model is encouraged to ask questions (by generating a few handcrafted ICL examples or formulaic interactions to FT on) highlight that the task being learned is perhaps not that complex.

**Justification For Why Not Lower Score:**

NA

---

### Decision · Program_Chairs · 2024-01-16

Reject